Proceedings of the 7th Symposium on Advances in Approximate Bayesian Inference, 2025 1–37

# $U$-ensembles: Improved diversity in the small data regime using unlabeled data

**Konstantinos Pitas**                                             KOSTASP210@GMAIL.COM
*Cognex Corporation, Switzerland*

**Hani Anouar Bourrous**                          HANI-ANOUAR.BOURROUS@INRIA.FR
**Julyan Arbel**                                                 JULYAN.ARBEL@INRIA.FR
*Univ. Grenoble Alpes, Inria, CNRS, Grenoble INP, LJK, 38000 Grenoble, France*

## Abstract

We present a method to improve the calibration of deep ensembles in the small data regime in the presence of unlabeled data. Our approach, which we name $U$-ensembles, is extremely easy to implement: given an unlabeled set, for each unlabeled data point, we simply fit a different randomly selected label with each ensemble member. We provide a theoretical analysis based on a PAC-Bayes bound which guarantees that for such a labeling we obtain low negative log-likelihood and high ensemble diversity on testing samples. Empirically, through detailed experiments, we find that for low to moderately-sized training sets, $U$-ensembles are more diverse and provide better calibration than standard ensembles.

## 1. Introduction

For many application settings of deep learning, there are abundant training data points, making it easy to train large well-calibrated networks. However, many real-world application settings remain in the small data regime, where training data is scarce and costly to sample and label (Perez-Ortiz et al., 2021, Bornschein et al., 2020, Foong et al., 2021), including in medical and industrial applications of deep learning. In such settings, predictors are often trained on limited datasets, ranging from several thousand to as few as a few dozen samples. In such settings, neural networks often operate in an auxiliary fashion to an expert. The network is expected to detect some positive samples, which might be missed by the expert, or the other way around. As such a successfully deployed system needs to be not only accurate but also well calibrated. The dominant approach in such settings is to fine-tune a pre-trained model on the few available data points. Alternatively, if relevant pre-trained models are not available, one has to resort to training a smaller architecture from scratch. In both cases, deep ensembles (Lakshminarayanan et al., 2017) and data augmentation (Shorten and Khoshgoftaar, 2019) play a key role in improving out-of-sample performance. To obtain well-calibrated predictions, tempering the final layer logits (Guo et al., 2017, Bornschein et al., 2020) is also crucial. Both empirically and theoretically, the performance of deep ensembles is intrinsically tied to their diversity (Fort et al., 2019, Masegosa, 2020). By averaging predictions from a more diverse set of models, prediction bias is mitigated, thereby enhancing overall performance.

The conventional approach to introducing diversity within deep ensembles involves employing distinct random initializations for each ensemble member (Lakshminarayanan et al., 2017). As a result, these ensemble members converge towards different modes of the loss landscape, each corresponding to a unique predictive function (Fort et al., 2019). This baseline technique is quite difficult to surpass. Nevertheless, numerous efforts have been made to

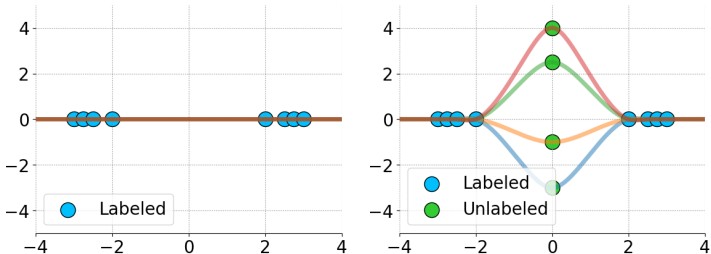

Figure 1: **Motivating $U$-ensembles.** (Left) Standard ensemble and (Right) $U$-ensemble on toy dataset. For the standard ensemble, all ensemble members converge to the same solution and the ensemble exhibits no diversity and hence no uncertainty in the "in-between" domain $[-2, 2]$. By fitting four different random labels for the unlabeled datapoint $x = 0$, $U$-ensembles learn four significantly different functions and therefore achieve diversity and informative uncertainty in the "in-between" domain.

further improve deep ensembles by explicitly encouraging diversity in their predictions (Ramé and Cord, 2021, Yashima et al., 2022, Masegosa, 2020, Pagliardini et al., 2023, D'Angelo and Fortuin, 2023). These approaches typically encounter several challenges, which can be summarized as follows: *The improvements in test metrics tend to be modest, while the associated extra costs are substantial.* Firstly, diversity-promoting algorithms often involve considerably more intricate implementation details compared to randomized initializations. Secondly, the computational and memory demands of existing methods exceed those of the baseline by a significant margin. Additionally, some approaches necessitate extensive hyperparameter tuning, further compounding computational costs.

In light of these considerations, we introduce $U$-ensembles, an algorithm designed to improve deep ensemble calibration and diversity with minimal deviations from the standard deep ensemble workflow. Moreover, our algorithm maintains the same computational and memory requirements as standard deep ensembles, resulting in linear increases in computational costs with the size of the unlabeled dataset. Our contributions are four-fold:

- Given an ensemble with $K$ members, a training set and an unlabeled set, we propose an algorithm, which we call $U$-ensembles, that generates for each unlabeled data point $K$ random labels and assigns from these a single random label to each ensemble member. For each ensemble member we then simply fit the training data (with its true labels) as well as the unlabeled data (with the generated random labels). See Figure 1.

- We provide a PAC-Bayesian analysis of the test performance of the proposed ensemble in terms of negative log-likelihood and diversity. The final ensemble is guaranteed to be diverse, and well-calibrated on test data.

- We train models from scratch on CIFAR-10 and CIFAR-100 for the in-distribution and out-of-distribution settings that demonstrate that for small to medium-sized training sets of the order of thousands of samples, $U$-ensembles improve the ECE by up to 1%.

- We explore the extremely small data regime of the order of dozens of training samples. We use ViT models pre-trained on Imagenet and fine-tune them on CIFAR-10 and CIFAR-100 and observe improvements in ECE of up to 12%.

Our method has similar computational requirements as standard ensembles and it only requires extending the training set with pseudo-labeled unlabeled datasets, thus it is ideally suited for AutoML (He et al., 2021) and EdgeML (Zhu et al., 2020) settings, where one has tight compute restrictions for fine-tuning and minimal access to the training objectives. Both settings overlap significantly with medical and industrial applications of deep learning (He et al., 2021, Zhu et al., 2020).

## 2. Works on deep ensemble improvements

The method closest to our approach is the Agree to Disagree algorithm (Pagliardini et al., 2023). Agree to Disagree also forces ensemble members to disagree with each other on unlabeled data. We deviate significantly from this work: i) Agree to Disagree ensembles are constructed greedily, ensemble members are added one at a time and are forced to disagree with the previous ones. By contrast, we propose an embarrassingly parallel optimization method; ii) In contrast to Pagliardini et al. (2023), we provide a detailed theoretical analysis on why promoting diversity on unlabeled data helps with generalization; iii) Empirically, we test the in-distribution setting which the original Agree to Disagree paper did not consider. We outperform Agree to Disagree ensembles in all experiments.

Wenzel et al. (2020) propose to induce diversity by training on different random initializations as well as different choices of hyperparameters such as the learning rate and the dropout rates in different layers. They also construct the ensemble greedily, yielding a memory complexity of either $\mathcal{O}(\#\text{random hyperparameter searches})$ or $\mathcal{O}((\text{ensemble size})^2)$. By contrast we only need $\mathcal{O}(\text{ensemble size})$ memory which is critical for EdgeML settings.

Ramé and Cord (2021), Yashima et al. (2022), Masegosa (2020), Ortega et al. (2022) all propose diversity promoting objectives. These methods exhibit all the shortcomings previously described, where the cost of implementation, tuning and training cannot easily be justified: 1) the implementation differs significantly from standard ensembles, 2) the computational complexity increases significantly, and 3) the algorithm requires extensive hyperparameter tuning.

D'Angelo and Fortuin (2023) propose to add a Kernelized repulsive term in the training update rule. This term discourages the ensemble members from converging towards the same function, thereby assuring members diversity. However this method presents several inherent challenges : 1) The choice and tuning of Kernel parameters (e.g., bandwidth) is not always straightforward. 2) Large ensembles with the kernelized term are more expensive to train compared to our method, due to the additional computational overhead.

Trinh et al. (2023) introduce First-order Repulsive Deep Ensembles (FoRDE), which adds a kernel-based repulsion term in input-gradient space so that networks within the ensemble learn complementary features. However, the approach has notable limitations: 1) Training is roughly three-times slower than standard ensembles. 2) Performance depends heavily on selecting suitable per-dimension kernel-length scales. 3) Computing and storing input gradients for every mini-batch increases memory use and limits scalability to very large models or datasets.

Jain et al. (2022), Lee et al. (2013) propose to pseudo-label unlabeled data using deep ensembles trained on labeled data. These pseudo-labeled data are then used to retrain the ensemble. This approach can improve significantly standard ensembles, however, it is significantly more costly than our method. First, unlabeled data have to be labeled in multiple rounds, a fraction at a time. Also, to be fully effective, ensembles have to be "distilled" into a final single network. Finally, diverse features have to be hand-designed based on the target task. Our method can be seen as the bare minimum version of pseudo-labeling with minimal assumptions on the generating data distribution.

Jain et al. (2022), Loh et al. (2023) encourage different ensemble members to be diverse using data transformations. These approaches are compatible with our method and could be used in conjunction with it. Note also the link of inducing diversity with the notion of functional priors on which there is considerable literature for Bayesian neural networks (Tran et al., 2022).

## 3. Diversity through unlabeled data

We consider the classification problem with $c$ classes. We denote by $1:K$ the set of integers $\{1, \ldots, K\}$, and let the training set be $Z = (X, Y) = \{(\boldsymbol{x}_i, y_i)\}_{i=1}^n \in (\mathcal{X} \times \mathcal{Y})^n$, drawn i.i.d from a distribution $\mathcal{D}$. Neural networks $f : \mathcal{X} \to \mathcal{Y}$ are parameterized by weights $\mathbf{w}$ and are denoted by $f = f(\cdot, \mathbf{w})$. Let $p(y|\boldsymbol{x}, f)$ be the probability of label $y$ given $\boldsymbol{x}$ and $f$. In our case $p(y|\boldsymbol{x}, f)$ is the softmax probability of class $y$ after the softmax function is applied to the logits of the last layer of a neural network. We consider sets of loss minima $\{\hat{\mathbf{w}}_i\}$ for $i \in 1:K$, and form a deep ensemble $\hat{\rho}(\mathbf{w}) = \frac{1}{K}\sum_i \delta(\mathbf{w} = \hat{\mathbf{w}}_i)$, where $\delta(x)$ is the Dirac delta mass function. To make a prediction on a new datapoint $(\boldsymbol{x}, y)$, the ensemble averages the probabilities estimated per class by each ensemble member

$$\hat{\boldsymbol{\mu}}(\hat{\rho}; (\boldsymbol{x}, y)) = \frac{1}{K}\sum_i p(y|\boldsymbol{x}, f(\boldsymbol{x}, \hat{\mathbf{w}}_i)), \tag{1}$$

with the typical goal of achieving low out-of-sample error $\mathbf{E}_{(\boldsymbol{x},y)\sim\mathcal{D}}\left[-\log\hat{\boldsymbol{\mu}}(\hat{\rho}; (\boldsymbol{x}, y))\right]$.

Let $\hat{\mathcal{L}}_Z^{\ell_{\mathrm{nll}}}(f) = -\frac{1}{n}\sum_i \log(p(y_i|\boldsymbol{x}_i, f))$ be the cross-entropy training loss. The standard deep ensemble algorithm then optimizes $\min_{\hat{\mathbf{w}}_i} \hat{\mathcal{L}}_Z^{\ell_{\mathrm{nll}}}(f(\cdot; \hat{\mathbf{w}}_i)) + \gamma\|\hat{\mathbf{w}}_i\|_2^2$ for each ensemble member $i \in 1 : K$ independently, using different random initializations for $\hat{\mathbf{w}}_i$. Even if all $\hat{\mathbf{w}}_i$ are learned independently, the different random initializations yield some diversity in the predictions. Instead of using only random initializations as a source of diversity, our method encourages the ensemble to be diverse on a new unlabeled set $U$. By learning useful features on the training set $Z$ and promoting diversity on the unlabeled set $U$, the resulting ensemble is more diverse than standard ensembles and generalizes better to new data. To extract meaningful features from $Z$, it is adequate to train with the real labels using the cross-entropy loss for each ensemble member, following the standard ensemble approach. Therefore, our focus is on finding an efficient method to ensure diversity within $U$.

**Promoting diversity on an unlabeled set $U$.** Given predictions $p(y|\boldsymbol{x}, f(\boldsymbol{x}, \hat{\mathbf{w}}_i))$ for each ensemble member $\hat{\mathbf{w}}_i$, we first need to define a metric of diversity which to be maximized. Specifically, we consider the empirical variance of $p(y|\boldsymbol{x}, f(\boldsymbol{x}, \hat{\mathbf{w}}_j))$ for ensemble $\hat{\rho}$, given a

signal $(\boldsymbol{x}, y)$, as a metric of diversity:

$$\hat{\mathbf{V}}(\hat{\rho}; (\boldsymbol{x}, y)) = \frac{1}{K} \sum_{i=1}^{K} \Big( p(y | \boldsymbol{x}, f(\boldsymbol{x}, \hat{\mathbf{w}}_i)) - \hat{\boldsymbol{\mu}}(\hat{\rho}; (\boldsymbol{x}, y)) \Big)^2. \tag{2}$$

The higher the variance, the higher the diversity of ensemble $\hat{\rho}$ on $(\boldsymbol{x}, y)$. Not only is this an intuitive diversity measure, but as established in Theorem 1, $\hat{\mathbf{V}}(\hat{\rho}; (\boldsymbol{x}, y))$ relates to a PAC-Bayesian analysis of the out-of-sample error.

We then make the crucial assumption that our ensemble can perfectly fit random labels[*] on unlabeled samples $(\boldsymbol{x}, y)$. This allows us to propose a very simple method to achieve a high value for the diversity term. Consider a single unlabeled point $(\boldsymbol{x}, y)$ in a two-class classification problem $y \in \{0, 1\}$, where the unknown label is $y = 1$, and two ensemble members $\hat{\mathbf{w}}_1$ and $\hat{\mathbf{w}}_2$ (here $y$ exists but is unknown during training). Let $\hat{\mathbf{w}}_1$ fit label 0 and $\hat{\mathbf{w}}_2$ fit label 1. Then $p(y = 1 | \boldsymbol{x}, \hat{\mathbf{w}}_1) = 0$ and $p(y = 1 | \boldsymbol{x}, \hat{\mathbf{w}}_2) = 1$ (because both ensemble members perfectly fit their assigned labels) and we get $\hat{\boldsymbol{\mu}}(\hat{\rho}; (\boldsymbol{x}, y = 1)) = 1/2$ and $\hat{\mathbf{V}}(\hat{\rho}; (\boldsymbol{x}, y = 1)) = 1/4$. Therefore, by assigning a distinct label to each ensemble member for every data point $(\boldsymbol{x}, y)$, we can achieve diverse predictions on the unlabeled set, even without prior knowledge of the true label $y = 1$ during training.

The most straightforward approach for generating diverse labels among ensemble members is to assign random labels. Given a $K$-ensemble, sampling $K$ labels from the $c$ classes for each unlabeled datapoint $\boldsymbol{x}$ can be done in two ways: **with replacement**, which can operate without constraint, or **without replacement**, which is applicable when $K \leq c$. We then assign one of the $K$ labels to each ensemble member. See Algorithm 1. In short, training $U$-ensembles simply requires to construct $K$ randomly labeled sets $U_i$ and then optimize

$$\hat{\mathcal{L}}_Z^{\ell_{\mathrm{nll}}}(f(\cdot; \hat{\mathbf{w}}_i)) + \beta \hat{\mathcal{L}}_{U_i}^{\ell_{\mathrm{nll}}}(f(\cdot; \hat{\mathbf{w}}_i)) + \gamma \|\hat{\mathbf{w}}_i\|_2^2 \tag{3}$$

with the optimization algorithm of our choice for each ensemble member.

---

**Algorithm 1** $U$-ensembles with or without replacement

---

**Input** Weight of the unlabeled loss $\beta$, $\ell_2$ regularization strength $\gamma$, training data $Z$, unlabeled data $U$, number of ensemble members $K$
**Output** Ensemble $\hat{\rho}(\mathbf{w}) = \frac{1}{K} \sum_i \delta(\mathbf{w} = \hat{\mathbf{w}}_i)$
1: **for** $i$ in $1 : K$ **do**
2:    $U_i \leftarrow \{\}$
3:    **for** $\boldsymbol{x}$ in $U$ **do**
4:        Sample $y$ with of without replacement from $1 : c$
5:        $U_i \leftarrow U_i \cup (\boldsymbol{x}, y)$
6:    **end for**
7:    $\hat{\mathbf{w}}_i \leftarrow$ Random Initialization
8:    $\min_{\hat{\mathbf{w}}_i} \hat{\mathcal{L}}_Z^{\ell_{\mathrm{nll}}}(f(\cdot; \hat{\mathbf{w}}_i)) + \beta \hat{\mathcal{L}}_{U_i}^{\ell_{\mathrm{nll}}}(f(\cdot; \hat{\mathbf{w}}_i)) + \gamma \|\hat{\mathbf{w}}_i\|_2^2$
9: **end for**

---

[*]The question of whether neural networks can fit random labels has been extensively explored in Zhang et al. (2021) who show that architectures as small as single-layer MLPs can easily fit random labels over CIFAR-10.

Table 1: **With replacement vs without replacement.** CIFAR-10 dataset and LeNet architecture. We analyze the case of in-distribution performance, 1000 training samples, 5000 unlabeled samples and 10 ensemble members. $U$-ensembles with replacement outperform standard ensembles and are in turn outperformed by $U$-ensembles without replacement.

| Method | Acc ↑ | ECE ↓ | TACE ↓ | Brier Rel. ↓ | NLL ↓ | $\mathbf{E}_{\hat{\rho}\sim\mathcal{A}}[\hat{\mathbf{V}}(\hat{\rho})]$ |
|---|---|---|---|---|---|---|
| Standard ensemble | 0.513 | 0.153 | 0.032 | 0.121 | 1.732 | - |
| $U$-ensemble w/ replacement | 0.510 | 0.137 | 0.030 | 0.120 | 1.680 | 0.04 |
| $U$-ensemble w/o replacement | **0.514** | **0.131** | **0.028** | **0.117** | **1.650** | 0.045 |

We derive now a PAC-Bayes bound[†] that links the out-of-sample loss to the diversity and the complexity of the classifier achieved by Algorithm 1 through random labeling.

**Theorem 1** *(Informal) Assume a training set $Z$ and an unlabeled set $U$ drawn from $\mathcal{D}$. Let Algorithm 1 generate an ensemble $\hat{\rho}(\mathbf{w}) = \frac{1}{K}\sum_i \delta(\mathbf{w} = \hat{\mathbf{w}}_i)$, which we denote by $\hat{\rho} \sim \mathcal{A}$, based on random labels $U$, where each member fits both real and random labels perfectly. With probability over $Z$ and $U$ larger than $1 - \delta$, for all $\gamma \in (0, 2)$ simultaneously, we have*

$$\mathbf{E}_{\hat{\rho}\sim\mathcal{A}}\big[\mathbf{E}_{(y,\boldsymbol{x})\sim\mathcal{D}}\big[-\log\sum_i p(y|\boldsymbol{x}, f(\cdot; \hat{\mathbf{w}}_i))/K\big]\big]$$

$$\leq \underbrace{\mathbf{E}_{\hat{\rho}\sim\mathcal{A}}\Big[\sum_i h_\delta\left(\|\hat{\mathbf{w}}_i\|_2^2\right)/K\Big]}_{\text{complexity}} - \underbrace{\left(1 - \frac{\gamma}{2}\right)\mathbf{E}_{\hat{\rho}\sim\mathcal{A}}\big[\hat{\mathbf{V}}(\hat{\rho}; U)\big]}_{\text{diversity}}$$

*where $h_\delta : \mathbb{R}^+ \to \mathbb{R}^+$ is a strictly increasing function, and $\hat{\mathbf{V}}(\hat{\rho}; U)$ is the variance term (2) of the ensemble on unlabeled data $U$.*

Theorem 1 indicates that if we can perfectly fit our training and unlabeled sets, out-of-sample performance is determined by a tradeoff between the ensemble complexity and the ensemble diversity on the unlabeled set $U$ (the result is in expectation over the randomness of the labels). Note that the loss on the training data does not appear on the RHS as it is 0 by definition, since we assume that we fit the labels perfectly.

However, even if we perfectly fit random labels on the unlabeled set, not all labellings result in the same diversity. We next find analytical values for the expected diversity of the ensemble generated by Algorithm 1.

**Proposition 2** *(Informal) If labels are assigned to the unlabeled data* with replacement *then*

$$\mathbf{E}_{\hat{\rho}\sim\mathcal{A}}\big[\hat{\mathbf{V}}(\hat{\rho})\big] = \frac{1}{2}\sum_{r=0}^{K} g(r)\binom{K}{r}\frac{1}{c^r}\left(1 - \frac{1}{c}\right)^{K-r}, \tag{4}$$

---

[†]Variants of the bound of Theorem 1 have appeared in recent works for majority-vote classifiers (Thiemann et al., 2017, Wu and Seldin, 2022, Masegosa et al., 2020, Masegosa, 2020). However, to the best of our knowledge, this particular version is novel in the deep ensemble case.

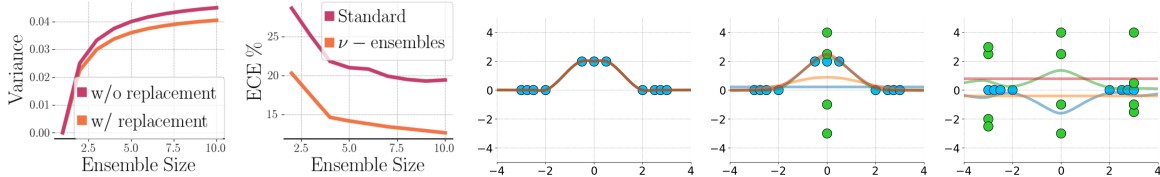

Figure 2: **$U$-ensembles and other methods.** Let $c = 10$ and let $K$ be an integer such that $1 \leq K \leq 10$ (i.e., $K \in 1 : 10$). (Top row) We plot $\mathbf{E}_{\hat{\rho} \sim \mathcal{A}}[\hat{\mathbf{V}}(\hat{\rho})]$ with and without replacement. Sampling without replacement results in more diverse ensembles. Improvements in ECE plateau around $K = 8$ for standard ensembles, but continue improving for $U$-ensembles. (Bottom left) Compared to Figure 1, with more training data the standard ensemble fits well the true function. (Bottom middle) The additional training data overlap with the unlabeled datapoint $x = 0$ and cause the $U$-ensemble to underfit. (Bottom right) The additional unlabeled datapoints $x = -3, x = 3$ overlap with training datapoints and cause the $U$-ensemble to underfit.

*where function $g$ is defined by*

$$g(r) = \frac{r}{K}\left(1 - \frac{r}{K}\right)^2 + (1 - \frac{r}{K})\left(\frac{r}{K}\right)^2.$$

*If labels are assigned to the unlabeled data* without replacement *then*

$$\mathbf{E}_{\hat{\rho} \sim \mathcal{A}}\left[\hat{\mathbf{V}}(\hat{\rho})\right] = \frac{1}{c}\left(1 - \frac{1}{K}\right). \tag{5}$$

The proof is deferred to Appendix B. We make three important observations from this analysis.

- $U$-ensembles achieve strictly increasing diversity with the ensemble size $K$ for both sampling with and without replacement (Equation (4), Equation (5) and Figure 2). This is not necessarily the case for standard ensembles.

- Equation (4), Equation (5) and Figure 2 indicate that sampling without replacement results in higher diversity compared to sampling with replacement.

- Our function, could become more complex without becoming sufficiently diverse. Then the complexity term in Theorem 1 will start to dominate the diversity term, resulting in worse out-of-sample performance. This can happen when $Z$ and $U$ start to overlap as we explain in Figure 2.

## 4. Experiments

We conducted four types of experiments: (i) small-scale experiments to test the intuitions from Section 3 (ii) experiments on in-distribution testing data when transfer learning isn't available, (iii) corresponding experiments on distribution shifts, (iv) experiments on transfer learning.

To approximate the presence of unlabeled data using common classification datasets, given a training set $Z$, we reserve a validation set $Z_{\text{val}}$, and a training set $Z_{\text{train}}$ and use the remaining datapoints as a pool for unlabeled data $U$. We keep the testing data $Z_{\text{test}}$ unchanged.

To test in-distribution performance, we use the standard CIFAR-10 and CIFAR-100 datasets (Krizhevsky and Hinton, 2009). We explore a variety of dataset sizes. Specifically, for both datasets, we keep the original testing set such that $|Z_{\text{test}}| = 10000$, and we use 5000 samples from the training set as unlabeled data $U$ and 5000 samples as validation data $Z_{\text{val}}$. For training, we use datasets $Z_{\text{train}}$ of size $1000, 2000, 4000, 10000$ and $40000$. We use three types of neural network architectures, a LeNet architecture (LeCun et al., 1998), an MLP architecture with 2 hidden layers (Goodfellow et al., 2016), and a WideResNet22 architecture (Zagoruyko and Komodakis, 2016). For both datasets, we used the standard augmentation setup of random flips + crops. We note that similar training-unlabeled set splits for CIFAR-10 and CIFAR-100 have been explored before in Alayrac et al. (2019), Jain et al. (2022).

We measure testing performance using accuracy as well as calibration on the testing set. Specifically, we measure calibration using the Expected Calibration Error (ECE) (Naeini et al., 2015), the Thresholded Adaptive Calibration Error (TACE) (Nixon et al., 2019), the Brier Score Reliability (Brier Rel.) (Murphy, 1973), and the Negative Log-Likelihood (NLL). We also measure the diversity of the ensemble on the test set using the average mutual information between ensemble member predictions. More specifically for each ensemble we treat its output as a random variable giving values in $1 : c$. We compute the Mutual Information (MI) of this random variable between all ensemble pairs and take the average. Lower MI then corresponds to more diverse ensembles.

For standard ensembles we simply minimize $\hat{\mathcal{L}}_Z^{\ell_{\text{nll}}}(f(\cdot; \hat{\mathbf{w}}_i)) + \gamma \|\hat{\mathbf{w}}_i\|_2^2$ for each ensemble member using different random initializations. For $U$-ensembles we optimize (3). For both cases, we train each ensemble member using AdamW (Loshchilov and Hutter, 2018). For hyperparameter tuning we perform a random search with 50 trials, using Hydra (Yadan, 2019). The details for the hyperparameter tuning ranges can be found in the supplementary. Table 2 presents the results for a training set of size 1000.

## 4.1. Small-scale experiments

We first establish that in accordance with Section 3, sampling without replacement outperforms sampling with replacement. We use an ensemble of size 10, the LeNet architecture and the CIFAR-10 dataset. In Table 1 we see that $U$-ensembles and sampling with replacement outperforms Standard ensembles in Expected Calibration Error by 1.6%. Sampling without replacement results in further improvements of 0.6%. We replicate these results on average with further experiments in the supplementary. *We thus use sampling without replacement in the rest of the experiments.* Furthermore, in Figure 2 we then show that increasing the ensemble size $K$ results in continuous improvements in ECE for the $U$-ensemble while the ECE for the Standard ensemble plateaus.

## 4.2. In-distribution performance

We present in Table 2 detailed results across our architectures for the CIFAR-100 dataset, where transfer learning is not applicable and full-model training is required. We also study the CIFAR-10 case, with results provided in Appendix E.

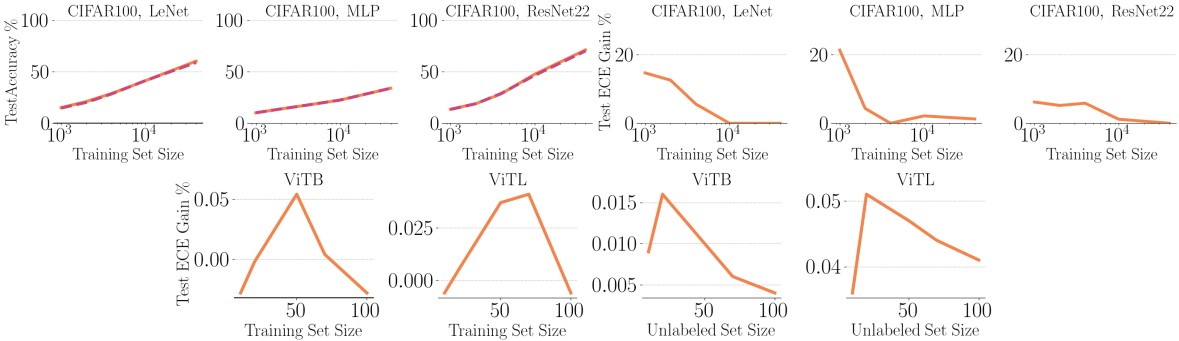

Figure 3: **Varying the training and unlabeled set.** (Top) For full-network training, we vary the labeled set size $Z \in \{1000, 2000, 4000, 10000, 40000\}$. $U$-ensembles match standard ensembles in accuracy (lines overlap) but consistently improve ECE, with diminishing gains as $Z$ grows. (Bottom) In ViT transfer learning, we fix one of $|Z| = 70$ or 100 and vary the other. Calibration initially improves but then degrades as the two sets grow and begin to overlap.

We see that $U$-ensembles have comparable accuracy to standard ensembles but with significantly better calibration across all calibration metrics. In terms of ECE we gain up to 12%. We also see that $U$-ensembles achieve significantly higher diversity between ensemble members by up to 0.4.

We also apply tempering (Guo et al., 2017) on Standard and $U$-ensembles, and see that $U$-ensembles retain gains in ECE of up to 1%. We use 3 seeds for all experiments.

We also compare with Masegosa ensembles (Masegosa, 2020), Agree to Disagree ensembles (Pagliardini et al., 2023) , Repulsive ensembles (D'Angelo and Fortuin, 2023) and Langevin dynamics (in the Appendix). We attempted to implement DICE ensembles (Ramé and Cord, 2021) but could not replicate a version that converged consistently. We see that Masegosa and Agree to Disagree ensembles tend to underfit the data and have worse testing accuracy than $U$-ensembles. In particular, Agree to Disagree ensembles also have in general worse calibration. Masegosa ensembles on the other hand have somewhat better calibration than $U$-ensembles in most cases. While repulsive ensembles show accuracy and MI on par with $U$-ensembles across different architectures, they underperform in terms of ECE. Additionally, we conducted experiments on Langevin ensembles, with the results reported in Appendix E.

During training, our algorithm compares very favorably in terms of time and space complexity with both Masegosa and Agree to Disagree ensembles. Standard and $U$ ensembles can have as low as constant memory cost as the ensemble size increases, if ensemble members are trained sequentially. On the other hand, Masegosa and Agree to Disagree ensembles in general scale like $\mathcal{O}(K)$ as all the ensemble members have to be trained jointly. Analyzing the computational cost is more complicated, however in general Masegosa ensembles require approximately $\times 2$ the computational time of Standard ensembles. Agree to Disagree ensembles scale roughly as $\mathcal{O}(K)$ as ensemble members have to be computed one at a time. In the supplementary we compare the computational cost of Standard, $U$ and Agree to Disagree ensembles.

Table 2: **In-distribution performance on CIFAR-100** (1000 training samples, 5000 unlabeled samples, 10 ensemble members). $U$-ensembles match standard ensembles in accuracy but outperform them in all calibration metrics, consistently across architectures. Their lower mutual information (MI) indicates greater diversity. Masegosa and Agree to Disagree often underfit, showing lower accuracy. We also assess the performance of $U$-ensembles in comparison with Repulsive ensembles : $U$-ensembles achieve comparable accuracy and MI but clearly outperform Repulsive ensembles in Expected Calibration Error (ECE). Tempering refers to (Guo et al., 2017), while $U$-Tempering combines this with U-ensembles.

| Dataset | Arch. | Method | Acc ↑ | ECE ↓ | TACE ↓ | Brier Rel. ↓ | NLL ↓ | MI ↓ |
|---------|-------|--------|-------|-------|--------|--------------|-------|------|
| CIFAR-100 | LeNet | Standard | **0.149** | 0.300 | 0.007 | 0.212 | 8.817 | 2.276 |
| | | Agree Dis. | 0.113 | 0.229 | 0.007 | 0.156 | 7.568 | 1.628 |
| | | Masegosa | 0.139 | 0.087 | 0.005 | 0.070 | 4.193 | 2.129 |
| | | Repulsive | 0.149 | 0.256 | 0.007 | 0.173 | 6.094 | 2.432 |
| | | Tempering | 0.149 | 0.017 | 0.0039 | 0.049 | **3.854** | 2.236 |
| | | **$U$-ensembles** | 0.147 | 0.186 | 0.006 | 0.131 | 5.115 | **1.826** |
| | | **$U$-Tempering** | 0.147 | **0.008** | 0.0038 | 0.048 | 3.929 | 1.661 |
| CIFAR-100 | MLP | Standard | 0.102 | 0.183 | 0.007 | 0.114 | 5.173 | 3.142 |
| | | Agree Dis. | 0.093 | 0.359 | 0.008 | 0.243 | 7.247 | 2.881 |
| | | Masegosa | 0.093 | 0.257 | 0.008 | 0.160 | 6.134 | 3.103 |
| | | Repulsive | 0.102 | 0.161 | 0.006 | 0.101 | 4.993 | 3.141 |
| | | Tempering | 0.102 | **0.008** | 0.00417 | **0.036** | 4.155 | 3.128 |
| | | **$U$-ensembles** | **0.103** | 0.156 | 0.006 | 0.106 | 4.906 | 3.014 |
| | | **$U$-Tempering** | 0.103 | 0.019 | 0.003 | 0.036 | **4.090** | **2.807** |
| CIFAR-100 | ResNet22 | Standard | **0.137** | 0.196 | 0.007 | 0.141 | 7.810 | 1.688 |
| | | Agree Dis. | 0.132 | 0.172 | 0.007 | 0.124 | 6.831 | 1.708 |
| | | Repulsive | 0.130 | 0.178 | 0.007 | 0.131 | 7.649 | 1.574 |
| | | Tempering | 0.137 | **0.011** | 0.004 | 0.040 | **3.891** | 1.608 |
| | | **$U$-ensembles** | 0.135 | 0.135 | 0.006 | 0.099 | 4.922 | 1.475 |
| | | **$U$-Tempering** | 0.135 | 0.018 | 0.003 | 0.036 | 3.930 | **1.432** |

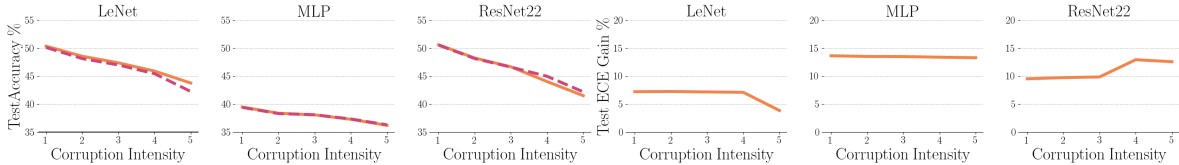

Figure 4: **CIFAR-10 robustness to common corruptions**. We evaluate our models on the CIFAR-10-C dataset (Hendrycks and Dietterich, 2018), which includes 15 corruption types at 5 intensity levels. For each level, we report the average test accuracy and ECE across corruptions. The $U$-ensemble matches the standard ensemble in accuracy and consistently achieves better calibration across all levels.

Table 3: **Transfer Learning Setup.** We explore an extremely small data regime using transfer learning from Imagenet to CIFAR-10 and CIFAR-100. $U$-ensembles improve the ECE by up to 12% over Standard ensembles.

| Dataset | Model | Ensemble | $Z$ | $U$ | Acc ↑ | ECE ↓ | TACE ↓ | Brier ↓ |
|---------|-------|----------|-----|-----|-------|-------|--------|---------|
| CIFAR-10 | ViTB16 | Standard | 20 | - | 0.629 | 0.101 | 0.057 | 0.516 |
| | | $U$-ensemble | | 20 | 0.628 | 0.022 | 0.055 | 0.496 |
| | ViTL16 | Standard | | - | 0.613 | 0.132 | 0.07 | 0.518 |
| | | $U$-ensemble | | 20 | 0.627 | 0.012 | 0.056 | 0.479 |
| CIFAR-100 | ViTB16 | Standard | 70 | - | 0.297 | 0.038 | 0.006 | 0.873 |
| | | $U$-ensemble | | 20 | 0.297 | 0.022 | 0.006 | 0.867 |
| | ViTL16 | Standard | | - | 0.339 | 0.077 | 0.006 | 0.802 |
| | | $U$-ensemble | | 20 | 0.345 | 0.026 | 0.006 | 0.785 |

Since the accuracy for CIFAR-100 is low with 1000 training samples, we then explore the effect of increasing the dataset size. We plot the results of varying the training set size in $\{1000, 2000, 4000, 10000, 40000\}$ in Figure 3. We observe that $U$-ensembles continue achieving the same accuracy as standard ensembles for all training set sizes. At the same time, they retain large improvements in calibration, in terms of the ECE, for small to medium size training sets. For larger training sets the improvements gradually decrease, as the training and unlabeled sets start to overlap. We obtain similar results for CIFAR-10.

### 4.3. Out-of-distribution (OOD) generalization

We evaluated $U$-ensembles and standard ensembles on difficult out-of-distribution tasks for CIFAR-10 dataset, for the case of 1000 training samples. Specifically, we followed the approach introduced in Hendrycks and Dietterich (2018) which proposed to evaluate the robustness of image classification algorithms to 15 common corruption types. We apply the corruption in 5 levels of increasing severity and evaluate the average test accuracy and calibration in terms of ECE across all corruption types. We plot the results Figure 4. We obeserve that $U$-ensembles retain the same testing accuracy as standard ensembles. At the same time, they are significantly better calibrated in terms of the Expected Calibration Error. This holds for all tested architectures and for all corruption levels. We further assess the robustness to Gaussian noise inputs; results are reported in Appendix G.

### 4.4. Transfer learning

We also explore the transfer learning setting where a classifier has been pretrained on data from a distribution $\mathcal{D}_1$ and fine-tuned for the target task on data from another distribution $\mathcal{D}_2$. Specifically, we use Vision Transformer (ViT) models that have been pretrained on Imagenet (Deng et al., 2009) and we fine-tune them on CIFAR-10 and CIFAR-100. We use two ViT architectures ViT-Base with 12 layers and ViT-Large with 24 layers, and an MLP fine-tuning head with 100 hidden dimensions. We explore settings with extremely small training set sizes $|Z| \in \{10, 20, 50, 70, 100\}$ and equivalently small unlabeled set sizes $|U| \in \{10, 20, 50, 70, 100\}$. We use ensembles of size 10 and perform the fine-tuning using AdamW. We compare Standard

ensembles and $U$-ensembles for $|Z| = 20$ for CIFAR-10 and $|Z| = 70$ for CIFAR-100 in Table 3, and defer all other results to the supplementary.

We see that even with these extremely small training sets the fine-tuned models learn non-trivial representations and achieve non-trivial test accuracy. $U$-ensembles achieve $\sim 10\%$ improvement in ECE on average for CIFAR-10 and $\sim 3.5\%$ improvement for CIFAR-100. Similarly the TACE, Brier improve consistently. Figure 3 plots the effect of varying the training and unlabeled set sizes, where ECE gains initially increase before gradually reversing. We further investigate transfer learning using ConvNeXt (Liu et al., 2022), SwinTransformer (Liu et al., 2021), and MaxViT (Tu et al., 2022) (all pretrained on ImageNet), where we fine-tune them on CIFAR-10, CIFAR-100, RxRx1 (Taylor et al., 2019), and iWildCam (Beery et al., 2020) under the same framework as ViT using ensembles of size 10 and AdamW; we defer the results to Appendix H.

## 5. Role of the hyperparameter $\beta$

We saw that if either the training or the unlabeled set is too large we might lose any gains in ECE, and other calibration metrics. In such a case standard ensembles could out-perform $U$-ensembles. In the presence of a validation set the hyperparameter $\beta$ plays a crucial role in ensuring good out-of-sample performance. By including $\beta = 0$ in a hyperparameter search we can always default to this value and therefore to Standard ensembles.

At the same time, in the small data regime, a validation set might not be available. In Figure 3 we observe that our gains show statistical regularity across different architectures. In the supplementary, we include additional experiments that show that our gains show statistical regularity *across different datasets as well*. This indicates that we should be able to assess whether unlabeled samples are beneficial solely through transfer learning. In particular, our experiments indicate that for datasets similar to CIFAR-10 and CIFAR-100 we should expect improvements from $U$-ensembles for $|Z| \approx 20$ and $|U| \approx 20$ while for other cases we should default to Standard ensembles.

## 6. Conclusion

In this work, we introduced $U$-ensembles, a novel deep ensemble algorithm that achieves improved calibration with minimal changes to the Standard deep ensemble pipeline, in the presence of unlabeled data. Future work includes circumventing the use of unlabeled data, by using augmented versions of existing datapoints, or even random Gaussian noise as inputs.

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

## Appendix A. Proof of Theorem 1

We start from the following Theorem by Masegosa, 2020.

**Theorem 1 (Masegosa, 2020)** *For any distribution $\hat{\rho}$ on $\mathcal{F}$*

$$\mathbf{E}_{(y,\boldsymbol{x})\sim\mathcal{D}}\left[-\ln\mathbf{E}_{\mathbf{w}\sim\hat{\rho}}\left[p(y|\boldsymbol{x},f(\boldsymbol{x};\mathbf{w}))\right]\right] \leq \mathbf{E}_{\mathbf{w}\sim\hat{\rho}}\left[\mathcal{L}_{(y,\boldsymbol{x})\sim\mathcal{D}}^{\ell_{\mathrm{nll}}}(f(\boldsymbol{x};\mathbf{w}))\right] - \mathbf{V}(\hat{\rho}) \tag{6}$$

*where $\mathbf{V}(\hat{\rho})$ is a variance term defined as*

$$\mathbf{V}(\hat{\rho}) = \mathbf{E}_{(y,\boldsymbol{x})\sim\mathcal{D}}\left[\frac{1}{2\max_{\mathbf{w}}p(y|\boldsymbol{x};\mathbf{w})}\mathbf{E}_{\mathbf{w}\sim\hat{\rho}}\left[(p(y|\boldsymbol{x},\mathbf{w})-\mathbf{E}_{\mathbf{w}\sim\hat{\rho}}\left(p(y|\boldsymbol{x},\mathbf{w}))\right)^2\right]\right]. \tag{7}$$

We need to bound $\mathbf{V}(\hat{\rho})$ and $\mathbf{E}_{\mathbf{w}\sim\hat{\rho}}\left[\mathcal{L}_{(y,\boldsymbol{x})\sim\mathcal{D}}^{\ell_{\mathrm{nll}}}(f(\boldsymbol{x};\mathbf{w}))\right]$ using their empirical versions. We will use a labeled training set $Z$ to bound the term $\mathbf{E}_{\mathbf{w}\sim\hat{\rho}}\left[\mathcal{L}_{(y,\boldsymbol{x})\sim\mathcal{D}}^{\ell_{\mathrm{nll}}}(f(\boldsymbol{x};\mathbf{w}))\right]$ and an unlabeled set $U$ to bound $\mathbf{V}(\hat{\rho})$. To bound the terms we will use existing PAC-Bayes bounds. The variance term has to be rewritten in the form $\mathbf{E}_{\mathbf{w}\sim\hat{\rho}}\mathbf{E}_{(y,\boldsymbol{x})\sim\mathcal{D}}\left[L(y,\boldsymbol{x},\mathbf{w})\right]$ in which PAC-Bayes bounds are directly applicable.

Let us assume as in Masegosa (2020) that the model likelihood is bounded:

**Assumption 2 (Masegosa, 2020)** *There exists a constant $C < \infty$ such that $\forall \boldsymbol{x} \in \mathcal{X}$, $\max_{y,\mathbf{w}} p(y|\boldsymbol{x};\mathbf{w}) \leq C$.*

Note that this assumption holds for the classification setting with $C = 1$. Then the variance can be written as

$$\begin{aligned}
\mathbf{V}(\hat{\rho}) &= \frac{1}{2}\mathbf{E}_{(y,\boldsymbol{x})\sim\mathcal{D}}\left[\mathbf{E}_{\mathbf{w}\sim\hat{\rho}}\left[(p(y|\boldsymbol{x},\mathbf{w})-\mathbf{E}_{\mathbf{w}\sim\hat{\rho}}\left(p(y|\boldsymbol{x},\mathbf{w}))\right)^2\right]\right] \\
&= \frac{1}{2}\mathbf{E}_{(y,\boldsymbol{x})\sim\mathcal{D}}\mathbf{E}_{\mathbf{w}\sim\hat{\rho}}\left[p(y|\boldsymbol{x},\mathbf{w})^2\right] - \frac{1}{2}\mathbf{E}_{(y,\boldsymbol{x})\sim\mathcal{D}}\left[\mathbf{E}_{\mathbf{w}\sim\hat{\rho}}p(y|\boldsymbol{x},\mathbf{w})\right]^2 \\
&= \frac{1}{2}\mathbf{E}_{(y,\boldsymbol{x})\sim\mathcal{D}}\mathbf{E}_{\mathbf{w}\sim\hat{\rho}}\left[p(y|\boldsymbol{x},\mathbf{w})^2\right] - \frac{1}{2}\mathbf{E}_{(y,\boldsymbol{x})\sim\mathcal{D}}\left[\mathbf{E}_{\mathbf{w}\sim\hat{\rho}}p(y|\boldsymbol{x},\mathbf{w})\mathbf{E}_{\mathbf{w}'\sim\hat{\rho}}p(y|\boldsymbol{x},\mathbf{w}')\right] \quad (8) \\
&= \frac{1}{2}\mathbf{E}_{(y,\boldsymbol{x})\sim\mathcal{D}}\mathbf{E}_{\hat{\rho}(\mathbf{w},\mathbf{w}')}\left[p(y|\boldsymbol{x},\mathbf{w})^2 - p(y|\boldsymbol{x},\mathbf{w})p(y|\boldsymbol{x},\mathbf{w}')\right] \\
&= \frac{1}{2}\mathbf{E}_{(y,\boldsymbol{x})\sim\mathcal{D}}\mathbf{E}_{\hat{\rho}(\mathbf{w},\mathbf{w}')}\left[L(y,\boldsymbol{x},\mathbf{w},\mathbf{w}')\right]
\end{aligned}$$

where $L(y,\boldsymbol{x},\mathbf{w},\mathbf{w}') = p(y|\boldsymbol{x},\mathbf{w})^2 - p(y|\boldsymbol{x},\mathbf{w})p(y|\boldsymbol{x},\mathbf{w}')$ and $\hat{\rho}(\mathbf{w},\mathbf{w}') = \hat{\rho}(\mathbf{w})\hat{\rho}(\mathbf{w}')$.

We can then use the following PAC-Bayes theorem to lower bound $\mathbf{V}(\hat{\rho})$ through it's empirical estimate, noting that $L(y,\boldsymbol{x},\mathbf{w},\mathbf{w}') \leq 1$ which is a requirement for this bound.

**Theorem 3 (PAC-Bayes-$\lambda$, [Thiemann et al., 2017](#))** *For any probability distribution $\pi$ on $\mathcal{F}$ that is independent of $U$ and any $\delta_1 \in (0,1)$, with probability at least $1 - \delta_1$ over a random draw of a sample $U$, for all distributions $\hat{\rho}$ on $\mathcal{F}$ and all $\gamma \in (0,2)$ simultaneously and a bounded loss $L \le 1$*

$$\mathbf{E}_{\mathbf{w}\sim\hat{\rho}}\mathbf{E}_{(y,\boldsymbol{x})\sim\mathcal{D}}\left[L(y,\boldsymbol{x},\mathbf{w})\right] \ge \left(1 - \frac{\gamma}{2}\right)\mathbf{E}_{\mathbf{w}\sim\hat{\rho}}\frac{1}{m}\sum_{(y,\boldsymbol{x})\in U}\left[L(y,\boldsymbol{x},\mathbf{w})\right] - \frac{\mathrm{KL}(\hat{\rho}||\pi) + \ln(2\sqrt{m}/\delta)}{\gamma m} \tag{9}$$

We then turn to the term $\mathbf{E}_{\mathbf{w}\sim\hat{\rho}}\left[\mathcal{L}_{(y,\boldsymbol{x})\sim\mathcal{D}}^{\ell_{\mathrm{nll}}}(f(\boldsymbol{x};\mathbf{w}))\right]$ where $L$ is unbounded due to the NLL loss. We will use the following bound:

**Theorem 4 ([Alquier et al., 2016](#))** *For any probability distribution $\pi$ on $\mathcal{F}$ that is independent of $Z$ and any $\delta_2 \in (0,1)$, with probability at least $1 - \delta_2$ over a random draw of a sample $Z$, for all distributions $\hat{\rho}$ on $\mathcal{F}$ and $\gamma > 0$*

$$\mathbf{E}_{\mathbf{w}\sim\hat{\rho}}\left[\mathcal{L}_{(y,\boldsymbol{x})\sim\mathcal{D}}^{\ell_{\mathrm{nll}}}(f(\boldsymbol{x};\mathbf{w}))\right] \le \mathbf{E}_{\mathbf{w}\sim\hat{\rho}}\left[\hat{\mathcal{L}}_Z^{\ell_{\mathrm{nll}}}(f(\boldsymbol{x};\mathbf{w}))\right] + \frac{\mathrm{KL}(\hat{\rho}||\pi) + \ln(\frac{1}{\delta}) + \psi_{\pi,\mathcal{D}}(\gamma,n)}{\gamma n} \tag{10}$$

*where*

$$\psi_{\pi,\mathcal{D}}(\gamma,n) = \ln\mathbf{E}_\pi\mathbf{E}_\mathcal{D}\left[e^{\gamma n\left(\mathcal{L}_{(y,\boldsymbol{x})\sim\mathcal{D}}^{\ell_{\mathrm{nll}}}(f(\boldsymbol{x};\mathbf{w})) - \hat{\mathcal{L}}_Z^{\ell_{\mathrm{nll}}}(f(\boldsymbol{x};\mathbf{w}))\right)}\right]. \tag{11}$$

By setting $\gamma_1 = \gamma_2 = \gamma/2$ and taking a union bound we then get:

**Theorem 5** *For any probability distribution $\pi$ on $\mathcal{F}$ that is independent of $U$ and $Z$ and any $\delta \in (0,1)$, with probability at least $1 - \delta$ over a random draw of a sample $U$ and $Z$, for all distributions $\hat{\rho}$ on $\mathcal{F}$ and all $\gamma \in (0,2)$ simultaneously*

$$\begin{aligned}
&\mathbf{E}_{(y,\boldsymbol{x})\sim\mathcal{D}}\left[-\ln\mathbf{E}_{\mathbf{w}\sim\hat{\rho}}\left[p(y|\boldsymbol{x},f(\boldsymbol{x};\mathbf{w}))\right]\right] \le \\
&\mathbf{E}_{\mathbf{w}\sim\hat{\rho}}\left[\hat{\mathcal{L}}_Z^{\ell_{\mathrm{nll}}}(f(\boldsymbol{x};\mathbf{w}))\right] + \frac{\mathrm{KL}(\hat{\rho}||\pi) + \ln(1/\delta) + \psi_{\pi,\mathcal{D}}(\gamma,n)}{\gamma n} \\
&- \left(1 - \frac{\gamma}{2}\right)\hat{\mathbf{V}}(\hat{\rho}) + \frac{\mathrm{KL}(\hat{\rho}||\pi) + \ln(2\sqrt{m}/\delta)}{\gamma m}.
\end{aligned} \tag{12}$$

What remains is to define the prior $\pi$ and posterior $\hat{\rho}$ distributions appropriately. We first set $\hat{\rho}(\mathbf{w}) = \frac{1}{K}\sum_i \delta(\mathbf{w} = \hat{\mathbf{w}}_i)$ which denotes an ensemble. We then follow [Masegosa (2020)](#) in properly defining the KL between $\hat{\rho}(\mathbf{w})$ and a given prior. Specifically, we restrict ourselves to a new family of priors, denoted $\pi_F(\mathbf{w})$. For any prior $\pi_F(\mathbf{w})$ within this family, its support is contained in $\mathbf{w}_F$, which denotes the space of real number vectors of dimension M that can be represented under a finite-precision scheme using F bits to encode each element of the vector. So we have $supp(\pi_F) \subseteq \mathbf{w}_F \subseteq \mathcal{R}^M$. This prior distribution $\pi_F$ can be expressed as, $\pi_F(\mathbf{w}) = \sum_{\mathbf{w}'\in\mathbf{w}_F} w_{\mathbf{w}'}\delta(\mathbf{w} = \mathbf{w}')$ where $w_{\mathbf{w}'}$ are positive scalar values parametrizing this prior distribution. They satisfy $w_{\mathbf{w}'} \ge 0$ and $\sum w_{\mathbf{w}'} = 1$. In this way, we can define a finite-precision counterpart to the Gaussian distribution where $w_{\mathbf{w}'} = \frac{1}{A}e^{-||\mathbf{w}'||_2^2}$ and $A$ is an

appropriate normalization constant. Puting everything back in (12) we get

$$
\mathbf{E}_{(y,\boldsymbol{x})\sim\mathcal{D}}\left[-\ln\frac{1}{K}\sum_i\left[p(y|\boldsymbol{x},f(\boldsymbol{x};\hat{\mathbf{w}}_i))\right]\right]
$$

$$
\leq \frac{1}{K}\sum_i\left[\hat{\mathcal{L}}_Z^{\ell_{\mathrm{nll}}}(f(\boldsymbol{x};\hat{\mathbf{w}}_i))\right] - \left(1-\frac{\gamma}{2}\right)\hat{\mathbf{V}}(\hat{\rho}) + \frac{1}{K}\sum_i h_\delta\left(\|\hat{\mathbf{w}}_i\|_2^2\right), \quad (13)
$$

where the increasing function $_\delta$ is defined by

$$
h_\delta\left(\|\hat{\mathbf{w}}_i\|_2^2\right) = \frac{\|\hat{\mathbf{w}}_i\|_2^2 + \ln A + K\ln(1/\delta) + K\psi_{\pi,\mathcal{D}}(\gamma,n)}{\gamma n} + \frac{\|\hat{\mathbf{w}}_i\|_2^2 + \ln A + K\ln(2\sqrt{m}/\delta)}{\gamma m}, \quad (14)
$$

which holds for any $\delta \in (0,1)$, with probability at least $1-\delta$ over a random draw of a sample $U$ and $Z$.

Given Inequality (13), we can take the expectation over an algorithm, $\hat{\rho}\sim\mathcal{A}$ which perfectly fits both real labels on the training set and random labels on the unlabeled set, to obtain

$$
\mathbf{E}_{\hat{\rho}\sim\mathcal{A}}\mathbf{E}_{(y,\boldsymbol{x})\sim\mathcal{D}}\left[-\ln\frac{1}{K}\sum_i\left[p(y|\boldsymbol{x},f(\boldsymbol{x};\hat{\mathbf{w}}_i))\right]\right]
$$

$$
\leq \mathbf{E}_{\hat{\rho}\sim\mathcal{A}}\left[\frac{1}{K}\sum_i\left[\hat{\mathcal{L}}_Z^{\ell_{\mathrm{nll}}}(f(\boldsymbol{x};\hat{\mathbf{w}}_i))\right]\right] - \mathbf{E}_{\hat{\rho}\sim\mathcal{A}}\left[\left(1-\frac{\gamma}{2}\right)\hat{\mathbf{V}}(\hat{\rho})\right] + \mathbf{E}_{\hat{\rho}\sim\mathcal{A}}\left[\frac{1}{K}\sum_i h_\delta\left(\|\hat{\mathbf{w}}_i\|_2^2\right)\right],
$$
$$(15)$$

which holds for any $\delta \in (0,1)$, with probability at least $1-\delta$ over a random draw of a sample $U$ and $Z$. We get Theorem 3.1 by noting that since $\mathcal{A}$ perfectly fits the training set then the first term on the RHS of (15) becomes zero.

Some further technical points need to be discussed at this point. Formally, Theorem 4 holds for a single value of $\gamma$. In order to combine both PAC-Bayes bounds we would need to form a grid over $\gamma$ in the range $(0,2)$ and do a union bound over this grid. The combined bound would then hold only for values on this grid. This results analysis only results in a negligible loosening of the bound (Dziugaite and Roy, 2017) and as such we neglect this discussion.

Since we have defined our bound in the discrete setting we cannot technically take derivatives of the resulting objective. However, as discussed in Masegosa (2020) during optimization we simply use the continuous version of all functions, knowing that we will arrive at a solution of finite precision.

## Appendix B. Proof of Proposition 2

We break this proof into two parts, sampling with replacement and sampling without replacement. We first note that the variance term has the following empirical form

$$
\hat{\mathbf{V}}(\hat{\rho}) = \frac{1}{2m}\sum_{(\boldsymbol{x},y)\in U}\left[\frac{1}{K}\sum_j\left[\left(p(y|\boldsymbol{x},\mathbf{w}_j) - \frac{1}{K}\sum_i\left(p(y|\boldsymbol{x},\mathbf{w}_i)\right)\right)^2\right]\right] \quad (16)
$$

we then explore this term in the two settings.

## B.1. Sampling without replacement

**Proposition 6** *Assume an unlabeled set $U \in \mathcal{D}^m$, $c$ number of classes, and a labeling distribution $\mathcal{R}$ which for each sample $(\boldsymbol{x}, \cdot) \in U$ selects $K \leq c$ labels from $(1 : c)$ randomly* ***without*** *replacement such that $\boldsymbol{y}_r \in (1 : c)^K$. Let $\mathcal{A}$ be an algorithm that takes $\boldsymbol{y}_r$ as input and generates an ensemble $\hat{\rho}(\mathbf{w}) = \frac{1}{K} \sum_i \delta(\mathbf{w} = \hat{\mathbf{w}}_i)$ such that $\forall i, f(\boldsymbol{x}, \hat{\mathbf{w}}_i)$ perfectly fits $\boldsymbol{y}_r[i]$*

$$\mathbf{E}_{\hat{\rho} \sim \mathcal{A}} \left[ \hat{\mathbf{V}}(\hat{\rho}) \right] = \frac{K - 1}{2cK} \tag{17}$$

*where the randomness is over $\boldsymbol{y}_r$ and we suppress the index for the different unlabeled points.*

**Proof**  We first discuss some preliminaries. We assume that each ensemble member fits the label assigned to it perfectly. Given a sample $(\boldsymbol{x}, y)$ and $K$ randomly sampled labels $\boldsymbol{y}_r \in (1 : c)^K$, without replacement, where only the $a$th label is the true label $y$, we have $p(y|\boldsymbol{x}, \mathbf{w}_a) = 1$ and $p(y|\boldsymbol{x}, \mathbf{w}_i) = 0, \forall i \neq a$.

The expectation of the variance term can now be simply obtained by separating the cases when $y$ is in the random labels $\boldsymbol{y}_r \in (1 : c)^K$ and the cases when it is not. We get

$$
\begin{aligned}
\mathbf{E}_{\hat{\rho} \sim \mathcal{A}} \left[ \hat{\mathbf{V}}(\hat{\rho}) \right] &= \mathbf{E}_{\hat{\rho} \sim \mathcal{A}} \left[ \frac{1}{2m} \sum_{(\boldsymbol{x}, y) \in U} \left[ \frac{1}{K} \sum_j \left[ \left( p(y|\boldsymbol{x}, \mathbf{w}_j) - \frac{1}{K} \sum_i (p(y|\boldsymbol{x}, \mathbf{w}_i)) \right)^2 \right] \right] \right] \\
&= \frac{1}{2m} \sum_U \left[ \frac{1}{K} \sum_j \left[ \left( p(y|\boldsymbol{x}, \mathbf{w}_j) - \frac{1}{K} \sum_i (p(y|\boldsymbol{x}, \mathbf{w}_i)) \right)^2 \right] \cdot \int \mathbb{I}\{y \text{ in randomized labels}\} dr \right. \\
&\quad \left. + \frac{1}{K} \sum_j \left[ \left( p(y|\boldsymbol{x}, \mathbf{w}_j) - \frac{1}{K} \sum_i (p(y|\boldsymbol{x}, \mathbf{w}_i)) \right)^2 \right] \cdot \int \mathbb{I}\{y \text{ not in randomized labels}\} dr \right] \\
&= \frac{1}{2m} \sum_U \left[ \frac{K - 1}{K^2} \cdot \int \mathbb{I}\{y \text{ in randomized labels}\} dr + 0 \cdot \int \mathbb{I}\{y \text{ not in randomized labels}\} dr \right] \\
&= \frac{1}{2m} \sum_U \frac{K - 1}{K^2} \cdot \frac{K}{c} \\
&= \frac{K - 1}{2cK}.
\end{aligned}
\tag{18}
$$

In line 4 we used the fact that the probability of sampling label $y$ in $K$ trials without replacement from a pool of $c$ labels is $\frac{K}{c}$.

In line 3 we used the fact that the term $\frac{1}{K} \sum_j \left[ \left( p(y|\boldsymbol{x}, \mathbf{w}_j) - \frac{1}{K} \sum_i (p(y|\boldsymbol{x}, \mathbf{w}_i)) \right)^2 \right]$ only has two possible values.

Let the true label $y$ be in the $K$ sampled labels, specifically let us assume that it is the $a$th sampled label. We can write

$$
\begin{aligned}
\frac{1}{K} \sum_j & \left[ \left( p(y|\boldsymbol{x}, \mathbf{w}_j) - \frac{1}{K} \sum_i (p(y|\boldsymbol{x}, \mathbf{w}_i)) \right)^2 \right] \\
&= \frac{1}{K} \sum_j \left[ \left( p(y|\boldsymbol{x}, \mathbf{w}_j) - \frac{1}{K} \left( p(y|\boldsymbol{x}, \mathbf{w}_a) + \sum_{i \neq a} p(y|\boldsymbol{x}, \mathbf{w}_i) \right) \right)^2 \right] \\
&= \frac{1}{K} \sum_j \left[ \left( p(y|\boldsymbol{x}, \mathbf{w}_j) - \frac{1}{K} (1+0) \right)^2 \right] \\
&= \frac{1}{K} \left( \left[ \left( p(y|\boldsymbol{x}, \mathbf{w}_a) - \frac{1}{K} \right)^2 \right] + \sum_{j \neq a} \left[ \left( p(y|\boldsymbol{x}, \mathbf{w}_j) - \frac{1}{K} \right)^2 \right] \right) \\
&= \frac{1}{K} \left( \left[ \left( 1 - \frac{1}{K} \right)^2 \right] + (K-1) \cdot \left[ \left( 0 - \frac{1}{K} \right)^2 \right] \right) \\
&= \frac{K-1}{K^2}.
\end{aligned}
\tag{19}
$$

Now let the true label $y$ *not* be in the $K$ sampled labels. We get

$$
\begin{aligned}
\frac{1}{K} \sum_j & \left[ \left( p(y|\boldsymbol{x}, \mathbf{w}_j) - \frac{1}{K} \sum_i (p(y|\boldsymbol{x}, \mathbf{w}_i)) \right)^2 \right] \\
&= \frac{1}{K} \sum_j \left[ \left( p(y|\boldsymbol{x}, \mathbf{w}_j) - \frac{1}{K} \cdot 0 \right)^2 \right] \\
&= \frac{1}{K} \sum_j \left[ (0 - 0)^2 \right] \\
&= 0,
\end{aligned}
\tag{20}
$$

where we use the fact that $p(y|\boldsymbol{x}, \mathbf{w}_i) = 0$, $\forall i$ if ensemble member $i$ does not fit the true label $y$ but another random label. ∎

## B.2. Sampling with replacement

Here we analyze the more complicated case of sampling with replacement. The crucial point is taking into account that the value of the variance term can be cast as the expectation of a function determined only by the number of times we draw the correct class $y$. We then use the fact that being successful $r$ times in $K$ independent trials with a probability $p = \frac{1}{c}$ of success corresponds to a Binomial distribution with parameters $K$ and $p = \frac{1}{c}$.

**Proposition 7** *Assume an unlabeled set $U \in \mathcal{D}^m$, $c$ number of classes, and a labeling distribution $\mathcal{R}$ which for each sample $(\boldsymbol{x}, \cdot) \in U$ selects $K \leq c$ labels from $(1 : c)$ randomly* **with** *replacement such that $\boldsymbol{y}_r \in (1 : c)^K$. Let $\mathcal{A}$ be an algorithm that takes $\boldsymbol{y}_r$ as input and generates an ensemble $\hat{\rho}(\mathbf{w}) = \frac{1}{K} \sum_i \delta(\mathbf{w} = \hat{\mathbf{w}}_i)$ such that $\forall i, f(\boldsymbol{x}, \hat{\mathbf{w}}_i)$ perfectly fits $\boldsymbol{y}_r[i]$*

$$\mathbf{E}_{\hat{\rho} \sim \mathcal{A}} \left[ \hat{\mathbf{V}}(\hat{\rho}) \right] = \frac{1}{2} \left[ \sum_r g(r) \binom{K}{r} \left( \frac{1}{c} \right)^r \left( 1 - \frac{1}{c} \right)^{K-r} \right] \tag{21}$$

*where $g(r) = \frac{1}{K} \left[ r \cdot \left( 1 - \frac{r}{K} \right)^2 + (K - r) \cdot \left( \frac{r}{K} \right)^2 \right]$, the randomness is over $\boldsymbol{y}_r$ and we suppress the index for the different unlabeled points.*

**Proof** To analyze this case we need to first assume that given a datasample $(\boldsymbol{x}, y)$ the value of $\frac{1}{K} \sum_j \left[ \left( p(y|\boldsymbol{x}, \mathbf{w}_j) - \frac{1}{K} \sum_i \left( p(y|\boldsymbol{x}, \mathbf{w}_i) \right) \right)^2 \right]$ only depends on $r$ the number of times we sample the true label $y$ in $K$ trials with replacement from a pool of $c$ possible labels. Let's then assume that the values are given from a function $g(r)$, it is obvious that what we are evaluating is the expectation of the function $g(r)$ under the Binomial distribution. We get

$$
\begin{aligned}
\mathbf{E}_{\hat{\rho} \sim \mathcal{A}} \left[ \hat{\mathbf{V}}(\hat{\rho}) \right] &= \mathbf{E}_{\hat{\rho} \sim \mathcal{A}} \left[ \frac{1}{2m} \sum_U \left[ \frac{1}{K} \sum_j \left[ \left( p(y|\boldsymbol{x}, \mathbf{w}_j) - \frac{1}{K} \sum_i \left( p(y|\boldsymbol{x}, \mathbf{w}_i) \right) \right)^2 \right] \right] \right] \\
&= \frac{1}{2m} \sum_U \left[ \sum_r g(r) \binom{K}{r} \left( \frac{1}{c} \right)^r \left( 1 - \frac{1}{c} \right)^{K-r} \right] \\
&= \frac{1}{2} \left[ \sum_r g(r) \binom{K}{r} \left( \frac{1}{c} \right)^r \left( 1 - \frac{1}{c} \right)^{K-r} \right]
\end{aligned}
\tag{22}
$$

where in line 3 we used the fact that the internal expectation is the same for all values in $U$.

To derive the form of $g(r)$ we first assume that given a sample $(\boldsymbol{x}, y)$ only $r$ out of $K$ trials with replacement sample the true label $y$. Denote the set of $r$ ensemble members that fit the true label $y$ as $S$. We then get

$$\frac{1}{K} \sum_j \left[ \left( p(y|\boldsymbol{x}, \mathbf{w}_j) - \frac{1}{K} \sum_i (p(y|\boldsymbol{x}, \mathbf{w}_i)) \right)^2 \right]$$

$$= \frac{1}{K} \sum_j \left[ \left( p(y|\boldsymbol{x}, \mathbf{w}_j) - \frac{1}{K} \left( \sum_{i \in S} p(y|\boldsymbol{x}, \mathbf{w}_i) + \sum_{i \notin S} p(y|\boldsymbol{x}, \mathbf{w}_i) \right) \right)^2 \right]$$

$$= \frac{1}{K} \sum_j \left[ \left( p(y|\boldsymbol{x}, \mathbf{w}_j) - \frac{1}{K} (r \cdot 1 + 0) \right)^2 \right]$$

$$= \frac{1}{K} \left( \sum_{j \in S} \left[ \left( p(y|\boldsymbol{x}, \mathbf{w}_j) - \frac{r}{K} \right)^2 \right] + \sum_{j \notin S} \left[ \left( p(y|\boldsymbol{x}, \mathbf{w}_j) - \frac{r}{K} \right)^2 \right] \right)$$

$$= \frac{1}{K} \left[ r \cdot \left( 1 - \frac{r}{K} \right)^2 + (K - r) \cdot \left( 0 - \frac{r}{K} \right)^2 \right]$$

$$= \frac{1}{K} \left[ r \cdot \left( 1 - \frac{r}{K} \right)^2 + (K - r) \cdot \left( \frac{r}{K} \right)^2 \right]$$

$$= g(r)$$

$$(23)$$

∎

## Appendix C. Additional conditions for a high-probability bound

Theorem 3.1 holds only in expectation. We give here some additional steps that are needed to obtain a high probability bound. Given inequality (13), we can take the expectation over an algorithm, $\hat{\rho} \sim \mathcal{A}$ which perfectly fits both real labels on the training set and random labels on the unlabeled set, to obtain

$$\mathbf{E}_{\hat{\rho} \sim \mathcal{A}} \mathbf{E}_{(y, \boldsymbol{x}) \sim \mathcal{D}} \left[ -\ln \frac{1}{K} \sum_i \left[ p(y|\boldsymbol{x}, f(\boldsymbol{x}; \hat{\mathbf{w}}_i)) \right] \right]$$

$$\leq \mathbf{E}_{\hat{\rho} \sim \mathcal{A}} \left[ \frac{1}{K} \sum_i \left[ \hat{\mathcal{L}}_Z^{\ell_{\mathrm{nll}}}(f(\boldsymbol{x}; \hat{\mathbf{w}}_i)) \right] \right] - \mathbf{E}_{\hat{\rho} \sim \mathcal{A}} \left[ \left( 1 - \frac{\gamma}{2} \right) \hat{\mathbf{V}}(\hat{\rho}) \right] + \mathbf{E}_{\hat{\rho} \sim \mathcal{A}} \left[ \frac{1}{K} \sum_i h_\delta \left( \|\hat{\mathbf{w}}_i\|_2^2 \right) \right],$$

$$(24)$$

which holds for any $\delta \in (0, 1)$, with probability at least $1 - \delta$ over a random draw of a sample $U$ and $Z$. We get Theorem 3.1 by noting that since $\mathcal{A}$ perfectly fits the training set then the first term on the RHS of (24) becomes zero.

First consider sampling without replacement such that we have

$$\mathbf{E}_{\hat{\rho} \sim \mathcal{A}} \left[ \left( 1 - \frac{\gamma}{2} \right) \hat{\mathbf{V}}(\hat{\rho}) \right] = \left( 1 - \frac{\gamma}{2} \right) \frac{K - 1}{2cK}.$$

Then, setting $L_1(\hat{\rho}) = \frac{1}{K} \sum_i \left[ \hat{\mathcal{L}}_Z^{\ell_{\mathrm{nll}}}(f(\boldsymbol{x}; \hat{\mathbf{w}}_i)) \right]$ and $L_2(\hat{\rho}) = \frac{1}{K} \sum_i h_\delta \left( \|\hat{\mathbf{w}}_i\|_2^2 \right)$ we note that both $L_1$ and $L_2$ are in general unbounded. To obtain a high-probability bound on

$$\mathbf{E}_{\hat{\rho} \sim \mathcal{A}} \mathbf{E}_{(y, \boldsymbol{x}) \sim \mathcal{D}} \left[ -\ln \frac{1}{K} \sum_i \left[ p(y | \boldsymbol{x}, f(\boldsymbol{x}; \hat{\mathbf{w}}_i)) \right] \right],$$

we need additional conditions on $\mathcal{A}$ namely that it outputs $\hat{\rho}$ such that $L_1(\hat{\rho}) \leq B$ and $L_2(\hat{\rho}) \leq C$ where $B, C$ are positive constants.

Then, for a finite sample $R \in \mathcal{A}^r$ and using Hoeffding's inequality and applying a union bound we can write

$$\mathbf{E}_{\hat{\rho} \sim \mathcal{A}} \mathbf{E}_{(y, \boldsymbol{x}) \sim \mathcal{D}} \left[ -\ln \frac{1}{K} \sum_{i \in \hat{\rho}} \left[ p(y | \boldsymbol{x}, f(\boldsymbol{x}; \hat{\mathbf{w}}_i)) \right] \right]$$

$$\leq \frac{1}{r} \sum_{\hat{\rho} \in R} \left[ \frac{1}{K} \sum_{i \in \hat{\rho}} \left[ \hat{\mathcal{L}}_Z^{\ell_{\mathrm{nll}}}(f(\boldsymbol{x}; \hat{\mathbf{w}}_i)) \right] \right] + \sqrt{\frac{B^2 \ln 1/b}{2r}}$$

$$- \left( 1 - \frac{\gamma}{2} \right) \frac{K-1}{2cK} + \frac{1}{r} \sum_{\hat{\rho} \in R} \left[ \frac{1}{K} \sum_{i \in \hat{\rho}} h_\delta \left( \|\hat{\mathbf{w}}_i\|_2^2 \right) \right] + \sqrt{\frac{C^2 \ln 1/c}{2r}},$$

which holds with probability $1 - (\delta + b + c)$ over the random draws of $U \in \mathcal{D}^m$, $Z \in \mathcal{D}^n$ and $R \in \mathcal{A}^r$ for $b, c \in (0, 1)$. The bound still holds *for the expectation* over $\hat{\rho} \sim \mathcal{A}$ and not with high probability for a single draw from $\mathcal{A}$. It guarantees that on average, ensembles that fit the training data and the randomly labeled data well, while having low complexity will generalize well to unseen data. In our experimental section, however, we have found that optimizing a single ensemble using our $U$-ensemble objective achieves all the desirable properties.

## Appendix D. Sampling with versus without replacement

We first plot the theoretical variance when sampling with replacement compared to when sampling without replacement. We see that sampling with replacement results in higher variance for the same number of ensemble members and thus higher diversity for the corresponding ensemble.

We perform the CIFAR-10 and CIFAR-100 experiments with an ensemble size of 10, using sampling with and without replacement and compare the results in Table 4. In the last row we compute the average difference between the metrics when sampling without replacement compared to sampling with replacement. We see that on average sampling without replacement results in improvements across different calibration metrics such as the ECE, Brier Reliability and Negative Log-Likelihood. The accuracy and the TACE remain relatively unchanged. At the same time the diversity of the ensemble also improves. These results validate our theoretical analysis, and further motivate improving the ensemble diversity using labels sampled without replacement.

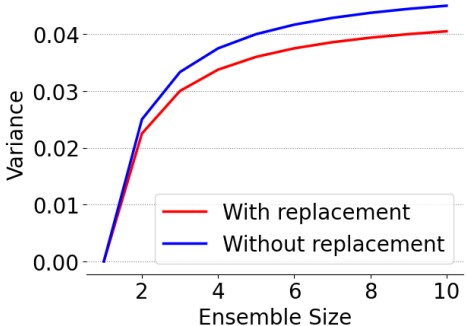

Figure 5: We consider $c = 10$ and $K \in 1 : 10$ and plot the variance term $\mathbf{E}_{\hat{\rho} \sim \mathcal{A}}[\hat{\mathbf{V}}(\hat{\rho})]$ with and without replacement. We see that sampling with replacement results in higher variance for the same number of ensemble members and thus higher diversity for the corresponding ensemble.

Table 4: **With replacement vs without replacement.** We analyze the case of ID performance, 1000 training samples, 10 ensemble members. Sampling the random labels in $U$-ensembles *without replacement*, on average results in improvements in calibration metrics such the ECE, the Brier Reliability and the Negative Log-Likelihood. The accuracy and the TACE remain relatively unchanged.

| Dataset / Arch. | Method | Acc ↑ | ECE ↓ | TACE ↓ | Brier Rel. ↓ | NLL ↓ | MI ↓ |
|---|---|---|---|---|---|---|---|
| CIFAR-10 | w/ replacement | 0.510 | 0.137 | 0.030 | 0.120 | 1.680 | **1.236** |
| / LeNet | w/o replacement | **0.514** | **0.131** | **0.028** | **0.117** | **1.650** | 1.245 |
| CIFAR-10 | w/ replacement | **0.401** | **0.083** | **0.022** | **0.087** | **1.753** | 1.650 |
| / MLP | w/o replacement | **0.401** | 0.098 | 0.023 | 0.092 | 1.767 | **1.559** |
| CIFAR-10 | w/ replacement | 0.520 | 0.016 | 0.020 | 0.090 | 1.471 | 0.699 |
| / ResNet22 | w/o replacement | **0.525** | **0.013** | **0.018** | **0.087** | **1.460** | **0.691** |
| CIFAR-100 | w/ replacement | 0.145 | 0.220 | **0.006** | 0.151 | 5.343 | 1.988 |
| / LeNet | w/o replacement | **0.147** | **0.155** | **0.006** | **0.113** | **4.846** | **1.654** |
| CIFAR-100 | w/ replacement | **0.103** | 0.116 | 0.005 | 0.078 | 4.447 | 3.047 |
| / MLP | w/o replacement | **0.103** | **0.040** | **0.004** | **0.049** | **4.171** | **2.807** |
| CIFAR-100 | w/ replacement | **0.134** | **0.093** | **0.006** | **0.074** | **4.266** | **1.086** |
| / ResNet22 | w/o replacement | **0.134** | 0.135 | **0.006** | 0.099 | 4.892 | 1.476 |
| mean diff. | w/o - w/ | **-0.001** | **-0.015** | **-0.0006** | **-0.010** | **-0.033** | **-0.031** |

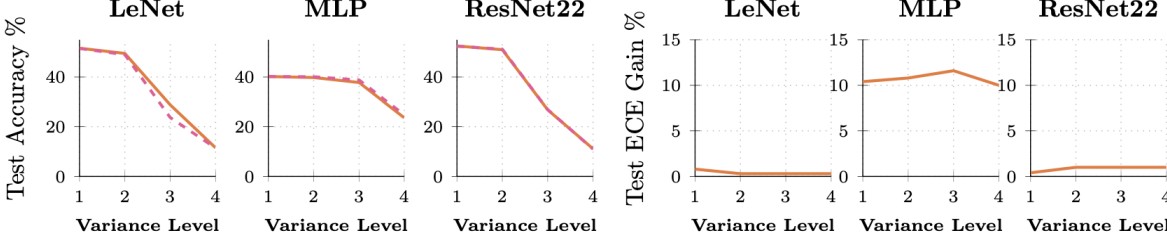

Figure 6: **CIFAR-10 robustness to Gaussian noise**. We evaluate our models on noisy CIFAR-10 images with different variance values. For each level, we report the average test accuracy and ECE across corruptions. We see that the $U$-ensembles have similar accuracy and yield better calibration in different settings.

## Appendix E. In-distribution performance

Table 5 shows the CIFAR-10 in-distribution performance for 1000 training samples, 5000 unlabeled samples and 10 ensemble members. $U$-ensembles yield similar accuracy to standard ensembles while reducing ECE by up to 11.9% and MI by up to 0.264. These improvements hold across all architectures. We also observe that the testing accuracy is low, however, this is to be expected due to the small size of the training dataset $Z_{\text{train}}$. We also apply tempering (Guo et al., 2017) on Standard and $U$-ensembles, and see that $U$-ensembles retain gains in ECE of up to 1%. We use 3 seeds for all experiments.

## Appendix F. Langevin dynamics experiments

We also compare against ensembles generated using Langevin dynamics. These are a popular and efficient method for obtaining diverse ensembles (Izmailov et al., 2021, Zhang et al., 2022, Garriga-Alonso and Fortuin, 2021), and are motivated as approximately sampling from the true Bayesian posterior.

For Langevin ensembles, we aim to minimize $\hat{\mathcal{L}}_Z^{\ell_{\text{nll}}}(f(\cdot; \hat{\mathbf{w}}_i)) + \gamma \|\hat{\mathbf{w}}_i\|_2^2$, with the following update rule $\mathbf{w}_{t+1} = \mathbf{w}_t - \alpha \nabla_{\mathbf{w}} \hat{\mathcal{L}}_Z^{\ell_{\text{nll}}}(f(\cdot; \mathbf{w}_t)) + \sqrt{2\alpha}\,\boldsymbol{\epsilon}$ where $\boldsymbol{\epsilon} \sim \mathcal{N}(0, \beta)$ and $\alpha$ represents the learning rate (Zhang et al., 2022, Garriga-Alonso and Fortuin, 2021). We ran the experiments on the LeNet and MLP architectures for both CIFAR-10 and CIFAR-100. We set $\beta = 1$ for the MLP and LeNet networks and $\beta = 0.001$ for the ResNet22 network. We used $\alpha = 0.001$ for all networks and trained for 300 epochs with a batch size of 32. The results are reported in Table 6.

## Appendix G. Robustness to random noise

We evaluated $U$-ensembles and standard ensembles on challenging out-of-distribution CIFAR-10 tasks using only 1,000 training samples, testing robustness to Gaussian noise with variances {0.001, 0.01, 0.1, 1} and reporting average test accuracy and ECE across corruptions in Figure 6, where $U$-ensembles match standard ensembles in accuracy while reducing ECE by 2% on LeNet and ResNet22 and 12% on MLP.

Table 5: **In-distribution performance on CIFAR-10** (1000 training samples, 5000 validation samples, 10 ensemble members). $U$-ensembles retain approximately the same accuracy as standard ensembles. At the same time, they achieve significantly better calibration in all calibration metrics. The improvements are consistent across all tested architectures and both datasets. We also observe that the Mutual Information (MI) of U-ensembles is significantly lower than standard ensembles. Thus, $U$-ensembles are more diverse than standard ensembles, which explains their improved calibration. Masegosa, Agree to Disagree, and Langevin ensembles often underfit, resulting in lower accuracy. Repulsive ensembles achieve strong accuracy and MI due to enforced diversity but have higher ECE than $U$-ensembles. Tempering follows Guo et al. (2017), while $U$-Tempering combines this approach with $U$-ensembles.

| Dataset | Arch. | Method | Acc ↑ | ECE ↓ | TACE ↓ | Brier Rel. ↓ | NLL ↓ | MI ↓ |
|---|---|---|---|---|---|---|---|---|
| CIFAR-10 | LeNet | Standard | **0.516** | 0.176 | 0.034 | 0.133 | 2.043 | 1.320 |
| | | Agree Dis. | 0.432 | 0.251 | 0.050 | 0.168 | 2.250 | 1.552 |
| | | Masegosa | 0.492 | 0.103 | 0.024 | **0.073** | 1.454 | **1.179** |
| | | Repulsive | 0.509 | 0.184 | 0.035 | 0.137 | 2.067 | 1.289 |
| | | Tempering | 0.516 | 0.024 | 0.0158 | 0.080 | **1.419** | 1.329 |
| | | $U$-ensembles | 0.511 | 0.133 | 0.028 | 0.118 | 1.664 | 1.201 |
| | | $U$-Tempering | 0.511 | **0.014** | **0.0145** | 0.080 | 1.437 | 1.215 |
| CIFAR-10 | MLP | Standard | 0.402 | 0.205 | 0.043 | 0.144 | 2.078 | 1.622 |
| | | Agree Dis. | 0.354 | 0.358 | 0.066 | 0.239 | 3.201 | 1.547 |
| | | Masegosa | 0.383 | 0.024 | 0.024 | 0.068 | 1.768 | 1.711 |
| | | Repulsive | **0.403** | 0.142 | 0.032 | 0.110 | 1.849 | 1.624 |
| | | Tempering | 0.402 | 0.020 | 0.0134 | **0.060** | 1.710 | 1.625 |
| | | $U$-ensembles | 0.401 | 0.086 | 0.023 | 0.087 | 1.782 | 1.525 |
| | | $U$-Tempering | 0.401 | **0.019** | **0.0133** | 0.060 | **1.690** | **1.554** |
| CIFAR-10 | ResNet22 | Standard | **0.527** | 0.087 | 0.024 | 0.106 | 1.690 | 0.939 |
| | | Agree Dis. | 0.478 | 0.051 | 0.020 | 0.087 | 1.633 | 0.706 |
| | | Repulsive | 0.509 | 0.092 | 0.025 | 0.107 | 1.704 | 0.905 |
| | | Tempering | 0.527 | 0.016 | **0.017** | **0.086** | **1.354** | 0.976 |
| | | $U$-ensembles | 0.527 | 0.014 | 0.017 | 0.082 | 1.436 | 0.675 |
| | | $U$-Tempering | 0.527 | **0.010** | **0.017** | **0.086** | 1.449 | **0.691** |

Table 6: **In-distribution performance on CIFAR-10 and CIFAR-100 for Langevin ensembles.** (1000 training samples, 5000 unlabeled samples, 10 ensemble members). $U$-Tempering combines tempering (Guo et al., 2017) with $U$-ensembles. We observe that Langevin ensembles underfit, resulting in lower accuracy compared to $U$-ensembles. Furthermore, we observe that the Mutual Information (MI) of U-ensembles is significantly lower than Langevin ensembles. Thus, $U$-ensembles are more diverse than Langevin ensembles. The severe underfitting of Langevin dynamics make comparison with U-ensembles difficult due to the known tradeoff between accuracy and calibratrion. In general U-ensembles with tempering outperform Langevin dynamics while also having higher accuracy.

| Dataset | Arch. | Method | Acc ↑ | ECE ↓ | TACE ↓ | Brier Rel. ↓ | NLL ↓ | MI ↓ |
|---------|-------|--------|-------|-------|--------|--------------|-------|------|
| CIFAR-10 | LeNet | Langevin | 0.424 | 0.047 | 0.016 | **0.073** | 1.625 | 1.824 |
| | | $U$-ensembles | **0.511** | 0.133 | 0.028 | 0.118 | 1.664 | **1.201** |
| | | $U$-Tempering | 0.511 | **0.014** | **0.0145** | 0.080 | **1.437** | 1.215 |
| | MLP | Langevin | 0.381 | 0.024 | 0.015 | **0.058** | 1.733 | 2.064 |
| | | $U$-ensembles | **0.401** | 0.086 | 0.023 | 0.087 | 1.782 | **1.525** |
| | | $U$-Tempering | 0.401 | **0.019** | **0.0133** | 0.060 | **1.690** | 1.554 |
| | ResNet22 | Langevin | 0.215 | 0.025 | 0.022 | **0.0** | 2.074 | 0.752 |
| | | $U$-ensembles | **0.527** | 0.014 | 0.017 | 0.082 | **1.436** | **0.675** |
| | | $U$-Tempering | 0.527 | **0.010** | **0.017** | 0.086 | 1.449 | **0.691** |
| CIFAR-100 | LeNet | Langevin | 0.114 | 0.146 | 0.006 | 0.094 | 4.595 | 2.972 |
| | | $U$-ensembles | **0.147** | 0.186 | 0.006 | 0.131 | 5.115 | 1.826 |
| | | $U$-Tempering | 0.147 | **0.008** | **0.0038** | **0.048** | **3.929** | **1.661** |
| | MLP | Langevin | 0.092 | **0.006** | 0.005 | **0.00** | 4.247 | 3.480 |
| | | $U$-ensembles | 0.103 | 0.156 | 0.006 | 0.106 | 4.906 | 3.014 |
| | | $U$-Tempering | **0.103** | 0.019 | **0.003** | 0.036 | **4.090** | **2.807** |
| | ResNet22 | Langevin | 0.033 | **0.010** | 0.004 | **0.00** | 4.408 | **0.859** |
| | | $U$-ensembles | **0.135** | 0.135 | 0.006 | 0.099 | 4.922 | 1.475 |
| | | $U$-Tempering | 0.135 | 0.018 | **0.003** | 0.036 | **3.930** | 1.432 |

## Appendix H. Transfer Learning

We evaluate our models in the transfer learning task by fine-tuning different classifiers on four datasets: CIFAR-10, CIFAR-100, RxRx1, and iWildCam. The last two datasets are part of the WILDS set of datasets (Koh et al., 2021) aimed at benchmarking out-of-distribution generalization. iWildCam contains images of animals, captured with camera traps as part of a wildlife biodiversity program. RxRx1 contains 3-channel images of cells obtained by fluorescent microscopy, and the label indicates which of 1,139 genetic treatments (including no treatment) the cells received.

Our experiments are conducted under an extremely low-data regime, using training sizes of $|Z| = 20$ for CIFAR-10 and $|Z| = 70$ for CIFAR-100, iWildCam. For all datasets, we include $|U| = 20$ unlabeled examples. In the case of RxRx1, due to its large number of classes, we set $|Z| = 500$ and $|U| = 100$, as smaller labeled sets lead to significantly low accuracy since RxRx1 has 1139 classes. All models are fine-tuned using AdamW, and we report results for

Table 7: **Transfer Learning Across Different Architectures and Datasets.** We conduct experiments to evaluate transfer learning from ImageNet to different datasets in the small-data regime. $U$-ensembles often show modest improvements in ECE across different architectures. .

| Dataset | $Z$ | $U$ | Model | Std-Acc ↑ | $U$-Acc ↑ | Std-ECE ↓ | $U$-ECE ↓ |
|---|---|---|---|---|---|---|---|
| CIFAR-10 | 20 | 20 | ConvNeXt | **0.4655** | 0.4471 | 0.079 | **0.058** |
| | | | SwinTransformer | 0.4217 | **0.4351** | 0.217 | **0.206** |
| | | | MaxViT | **0.4294** | 0.3479 | 0.123 | **0.033** |
| CIFAR-100 | 70 | 20 | ConvNeXt | 0.1688 | **0.1693** | 0.009 | 0.009 |
| | | | SwinTransformer | 0.1783 | **0.1820** | **0.115** | 0.121 |
| | | | MaxViT | 0.1167 | **0.1253** | 0.053 | **0.045** |
| RxRx1 | 500 | 100 | ConvNeXt | 0.0061 | **0.0064** | **0.0121** | 0.0123 |
| | | | SwinTransformer | 0.0042 | **0.0048** | 0.008 | **0.0065** |
| | | | MaxViT | 0.0077 | **0.0080** | **0.0022** | 0.0025 |
| iWildCam | 70 | 20 | ConvNeXt | **0.3399** | 0.3384 | 0.108 | **0.094** |
| | | | SwinTransformer | **0.3800** | 0.3090 | **0.078** | 0.085 |
| | | | MaxViT | **0.3307** | 0.3300 | 0.313 | **0.298** |

ensembles of size 10. A comparison between standard ensembles and $U$-ensembles is given in Table 7.

Even with extremely small training sets, the fine-tuned models learn non-trivial representations and achieve meaningful test accuracy. $U$-ensembles often outperform standard ensembles in terms of ECE for various architectures, while retaining the standard ensemble accuracy.

## Appendix I. Additional datasets

Here we explore additional datasets, the SVHN dataset (Buitinck et al., 2013) and the STL10 dataset (Coates et al., 2011). We use 1000 training samples, 3000 validation samples, 1000 unlabeled samples and the original test sets for both datasets. We plot the results in Table 8. We see that on average the results much those for the CIFAR-10 and CIFAR-100 cases. $U$-ensembles achieve improvements in calibration while typically not hurting accuracy.

## Appendix J. Experimental setup

We ran all experiments using A100, and V100 NVIDIA GPUs on our cluster. In total, the experiments consumed approximately 10000 hours of GPU time. The implementations were done in JAX (Bradbury et al., 2018). While data loading was done in Tensorflow (Abadi et al., 2015). For $U$-ensembles, for the LeNet architecture we investigated epochs in the range $[100, 120, 140, 160, 180, 200, 220, 240, 260]$, for the MLP $[100, 120, 140, 160, 180, 200, 220, 240, 260]$, for the ResNet $[200, 220, 250, 270, 300, 320, 350, 370, 400]$. For the regularization strength, we searched in the range $[1, 0.1, 0.05, 0.01, 0]$ and for the optimizer learning rate

Table 8: **ID performance, 1000 training samples, 5000 unlabeled samples, 10 ensemble members.** $U$-ensembles retain approximately the same accuracy as standard ensembles. At the same time, they achieve significantly better calibration in all calibration metrics. These results are consistent with the experiments for the CIFAR-10 and the CIFAR-100. The only outlier is the ResNet22 architecture for the STL10 dataset, where $U$-ensembles underfit the data.

| Dataset / Aug | Method | Acc ↑ | ECE ↓ | TACE ↓ | Brier Rel. ↓ | NLL ↓ | MI ↓ |
|---|---|---|---|---|---|---|---|
| SVHN | Standard | **0.618** | 0.138 | 0.028 | 0.114 | 1.679 | **1.561** |
| / LeNet | $\nu$-ensembles | 0.605 | **0.083** | **0.023** | **0.102** | **1.371** | 1.792 |
| SVHN | Standard | **0.474** | 0.252 | 0.047 | 0.170 | 2.748 | 1.733 |
| / MLP | $\nu$-ensembles | 0.471 | **0.157** | **0.037** | **0.128** | **2.008** | **1.653** |
| SVHN | Standard | **0.707** | **0.070** | **0.019** | 0.098 | 1.012 | 0.921 |
| / ResNet22 | $\nu$-ensembles | 0.700 | **0.070** | **0.019** | **0.096** | **0.988** | **0.906** |
| STL10 | Standard | 0.309 | 0.045 | 0.018 | 0.051 | 1.91 | **1.821** |
| / LeNet | $\nu$-ensembles | **0.310** | **0.020** | **0.017** | **0.043** | **1.896** | 1.854 |
| STL10 | Standard | **0.302** | 0.021 | 0.016 | 0.037 | 1.905 | 1.714 |
| / MLP | $\nu$-ensembles | **0.302** | **0.013** | **0.015** | **0.033** | **1.897** | **1.660** |
| STL10 | Standard | **0.302** | **0.037** | **0.018** | **0.045** | **1.898** | **0.865** |
| / ResNet22 | $\nu$-ensembles | 0.278 | 0.217 | 0.050 | 0.134 | 2.423 | 0.642 |

Table 9: Full results for CIFAR-10 ViTB16 fine-tuning experiments.

| Model | Ensemble | $Z$ Size | $U$ Size | Acc ↑ | ECE ↓ | TACE ↓ | Brier ↓ | AUROC ↑ |
|-------|----------|----------|----------|-------|-------|--------|---------|---------|
| ViTB16 | standard | 10 | 10 | 0.39 | 0.093 | 0.094 | 0.766 | 0.802 |
| ViTB16 | standard | 20 | 10 | 0.629 | 0.101 | 0.057 | 0.516 | 0.856 |
| ViTB16 | standard | 50 | 10 | 0.873 | 0.025 | 0.019 | 0.188 | 0.877 |
| ViTB16 | standard | 70 | 10 | 0.888 | 0.01 | 0.013 | 0.162 | 0.902 |
| ViTB16 | $U$-ensembles | 10 | 10 | 0.472 | 0.036 | 0.069 | 0.677 | 0.791 |
| ViTB16 | $U$-ensembles | 10 | 50 | 0.452 | 0.03 | 0.048 | 0.659 | 0.839 |
| ViTB16 | $U$-ensembles | 10 | 100 | 0.458 | 0.052 | 0.069 | 0.681 | 0.826 |
| ViTB16 | $U$-ensembles | 20 | 10 | 0.655 | 0.042 | 0.059 | 0.495 | 0.823 |
| ViTB16 | $U$-ensembles | 20 | 20 | 0.628 | 0.022 | 0.055 | 0.496 | 0.856 |
| ViTB16 | $U$-ensembles | 20 | 50 | 0.646 | 0.032 | 0.053 | 0.483 | 0.861 |
| ViTB16 | $U$-ensembles | 20 | 70 | 0.637 | 0.044 | 0.052 | 0.479 | 0.872 |
| ViTB16 | $U$-ensembles | 20 | 100 | 0.625 | 0.075 | 0.064 | 0.506 | 0.86 |
| ViTB16 | $U$-ensembles | 50 | 10 | 0.867 | 0.063 | 0.022 | 0.208 | 0.864 |
| ViTB16 | $U$-ensembles | 50 | 20 | 0.879 | 0.095 | 0.025 | 0.199 | 0.866 |
| ViTB16 | $U$-ensembles | 50 | 50 | 0.877 | 0.089 | 0.025 | 0.197 | 0.872 |
| ViTB16 | $U$-ensembles | 50 | 70 | 0.871 | 0.087 | 0.026 | 0.204 | 0.87 |
| ViTB16 | $U$-ensembles | 50 | 100 | 0.862 | 0.104 | 0.028 | 0.222 | 0.867 |
| ViTB16 | $U$-ensembles | 70 | 10 | 0.878 | 0.038 | 0.017 | 0.183 | 0.895 |
| ViTB16 | $U$-ensembles | 70 | 20 | 0.864 | 0.053 | 0.021 | 0.205 | 0.882 |
| ViTB16 | $U$-ensembles | 70 | 50 | 0.885 | 0.075 | 0.024 | 0.18 | 0.896 |
| ViTB16 | $U$-ensembles | 70 | 70 | 0.886 | 0.075 | 0.023 | 0.178 | 0.897 |
| ViTB16 | $U$-ensembles | 70 | 100 | 0.852 | 0.106 | 0.03 | 0.232 | 0.882 |

in $[0.0001, 0.001]$. We investigated the same epoch and learning rate ranges for Standard ensembles. Agree to Disagree ensembles contain a single hyperparameter $\alpha$. We tested values in the range $[1, 0.1, 0.01, 0.001, 0.0001]$.

For the transfer learning experiments, we used the ViT implementations available in Torchvision (Marcel and Rodriguez, 2010). These were pre-trained on Imagenet.

## Appendix K. Additional figures

In Figure 7 we plot the effect of varying the training set and the unlabeled set. In Figure 8 we make some comparisons between the computational time for $U$-ensembles vs other methods. In Tables 9, 10, 11, 12 we plot the full results on the fine-tuning experiments for CIFAR-10 and CIFAR-100.

Table 10: Full results for CIFAR-10 ViTL16 fine-tuning experiments.

| Model | Ensemble | $Z$ Size | $U$ Size | Acc ↑ | ECE ↓ | TACE ↓ | Brier ↓ | AUROC ↑ |
|---|---|---|---|---|---|---|---|---|
| ViTL16 | standard | 10 | 10 | 0.44 | 0.095 | 0.091 | 0.697 | 0.839 |
| ViTL16 | standard | 20 | 10 | 0.613 | 0.132 | 0.07 | 0.518 | 0.931 |
| ViTL16 | standard | 50 | 10 | 0.936 | 0.025 | 0.009 | 0.098 | 0.928 |
| ViTL16 | standard | 70 | 10 | 0.931 | 0.01 | 0.01 | 0.103 | 0.926 |
| ViTL16 | $U$-ensembles | 10 | 10 | 0.497 | 0.027 | 0.073 | 0.631 | 0.848 |
| ViTL16 | $U$-ensembles | 10 | 50 | 0.497 | 0.09 | 0.064 | 0.61 | 0.913 |
| ViTL16 | $U$-ensembles | 10 | 100 | 0.482 | 0.069 | 0.075 | 0.641 | 0.874 |
| ViTL16 | $U$-ensembles | 20 | 10 | 0.618 | 0.028 | 0.067 | 0.504 | 0.882 |
| ViTL16 | $U$-ensembles | 20 | 20 | 0.627 | 0.012 | 0.056 | 0.479 | 0.894 |
| ViTL16 | $U$-ensembles | 20 | 50 | 0.631 | 0.041 | 0.053 | 0.46 | 0.907 |
| ViTL16 | $U$-ensembles | 20 | 70 | 0.644 | 0.048 | 0.052 | 0.446 | 0.913 |
| ViTL16 | $U$-ensembles | 20 | 100 | 0.648 | 0.073 | 0.048 | 0.439 | 0.915 |
| ViTL16 | $U$-ensembles | 50 | 10 | 0.924 | 0.062 | 0.015 | 0.133 | 0.895 |
| ViTL16 | $U$-ensembles | 50 | 20 | 0.929 | 0.064 | 0.016 | 0.117 | 0.918 |
| ViTL16 | $U$-ensembles | 50 | 50 | 0.933 | 0.066 | 0.015 | 0.112 | 0.92 |
| ViTL16 | $U$-ensembles | 50 | 70 | 0.93 | 0.079 | 0.018 | 0.121 | 0.918 |
| ViTL16 | $U$-ensembles | 50 | 100 | 0.918 | 0.102 | 0.024 | 0.148 | 0.903 |
| ViTL16 | $U$-ensembles | 70 | 10 | 0.929 | 0.038 | 0.012 | 0.118 | 0.907 |
| ViTL16 | $U$-ensembles | 70 | 20 | 0.93 | 0.063 | 0.017 | 0.117 | 0.91 |
| ViTL16 | $U$-ensembles | 70 | 50 | 0.922 | 0.051 | 0.017 | 0.123 | 0.912 |
| ViTL16 | $U$-ensembles | 70 | 70 | 0.927 | 0.058 | 0.016 | 0.12 | 0.913 |
| ViTL16 | $U$-ensembles | 70 | 100 | 0.923 | 0.078 | 0.02 | 0.133 | 0.905 |

Table 11: Full results for CIFAR-100 ViTB16 fine-tuning experiments.

| Model | Ensemble | *Z* Size | *U* Size | Acc ↑ | ECE ↓ | TACE ↓ | Brier ↓ | AUROC ↑ |
|-------|----------|----------|----------|-------|-------|--------|---------|---------|
| ViTB16 | standard | 10 | 10 | 0.076 | 0.037 | 0.007 | 0.984 | 0.907 |
| ViTB16 | standard | 20 | 10 | 0.131 | 0.065 | 0.009 | 0.97 | 0.829 |
| ViTB16 | standard | 50 | 10 | 0.231 | 0.079 | 0.007 | 0.926 | 0.78 |
| ViTB16 | standard | 70 | 10 | 0.297 | 0.038 | nan | 0.873 | 0.766 |
| ViTB16 | *U*-ensembles | 10 | 10 | 0.077 | 0.037 | 0.007 | 0.984 | 0.893 |
| ViTB16 | *U*-ensembles | 10 | 50 | 0.076 | 0.038 | 0.007 | 0.984 | 0.914 |
| ViTB16 | *U*-ensembles | 10 | 100 | 0.076 | 0.065 | 0.008 | 0.976 | 0.911 |
| ViTB16 | *U*-ensembles | 20 | 10 | 0.129 | 0.065 | 0.009 | 0.971 | 0.829 |
| ViTB16 | *U*-ensembles | 20 | 20 | 0.126 | 0.053 | 0.009 | 0.97 | 0.819 |
| ViTB16 | *U*-ensembles | 20 | 50 | 0.132 | 0.053 | 0.011 | 0.962 | 0.844 |
| ViTB16 | *U*-ensembles | 20 | 70 | 0.135 | 0.056 | 0.01 | 0.96 | 0.848 |
| ViTB16 | *U*-ensembles | 20 | 100 | 0.134 | 0.067 | 0.011 | 0.96 | 0.853 |
| ViTB16 | *U*-ensembles | 50 | 10 | 0.235 | 0.028 | nan | 0.913 | 0.787 |
| ViTB16 | *U*-ensembles | 50 | 20 | 0.215 | 0.039 | nan | 0.92 | 0.82 |
| ViTB16 | *U*-ensembles | 50 | 50 | 0.234 | 0.061 | 0.007 | 0.922 | 0.773 |
| ViTB16 | *U*-ensembles | 50 | 70 | 0.238 | 0.037 | nan | 0.91 | 0.788 |
| ViTB16 | *U*-ensembles | 50 | 100 | 0.233 | 0.025 | 0.01 | 0.917 | 0.765 |
| ViTB16 | *U*-ensembles | 70 | 10 | 0.288 | 0.029 | nan | 0.879 | 0.762 |
| ViTB16 | *U*-ensembles | 70 | 20 | 0.297 | 0.022 | nan | 0.867 | 0.768 |
| ViTB16 | *U*-ensembles | 70 | 50 | 0.296 | 0.028 | nan | 0.869 | 0.763 |
| ViTB16 | *U*-ensembles | 70 | 70 | 0.298 | 0.032 | nan | 0.869 | 0.764 |
| ViTB16 | *U*-ensembles | 70 | 100 | 0.293 | 0.034 | nan | 0.874 | 0.768 |

Table 12: Full results for CIFAR-100 ViTL16 fine-tuning experiments.

| Model | Ensemble | $Z$ Size | $U$ Size | Acc ↑ | ECE ↓ | TACE ↓ | Brier ↓ | AUROC ↑ |
|---|---|---|---|---|---|---|---|---|
| ViTL16 | standard | 10 | 10 | 0.08 | 0.042 | 0.007 | 0.981 | 0.899 |
| ViTL16 | standard | 20 | 10 | 0.136 | 0.051 | 0.01 | 0.951 | 0.873 |
| ViTL16 | standard | 50 | 10 | 0.25 | 0.056 | 0.012 | 0.89 | 0.831 |
| ViTL16 | standard | 70 | 10 | 0.339 | 0.077 | 0.006 | 0.802 | 0.869 |
| ViTL16 | $U$-ensembles | 10 | 10 | 0.074 | 0.042 | 0.008 | 0.981 | 0.904 |
| ViTL16 | $U$-ensembles | 10 | 50 | 0.086 | 0.045 | 0.008 | 0.964 | 0.904 |
| ViTL16 | $U$-ensembles | 10 | 100 | 0.087 | 0.048 | 0.009 | 0.965 | 0.892 |
| ViTL16 | $U$-ensembles | 20 | 10 | 0.137 | 0.053 | 0.009 | 0.951 | 0.871 |
| ViTL16 | $U$-ensembles | 20 | 20 | 0.139 | 0.046 | 0.01 | 0.947 | 0.863 |
| ViTL16 | $U$-ensembles | 20 | 50 | 0.144 | 0.04 | 0.01 | 0.942 | 0.853 |
| ViTL16 | $U$-ensembles | 20 | 70 | 0.143 | 0.041 | 0.01 | 0.94 | 0.864 |
| ViTL16 | $U$-ensembles | 20 | 100 | 0.139 | 0.046 | 0.01 | 0.948 | 0.864 |
| ViTL16 | $U$-ensembles | 50 | 10 | 0.258 | 0.063 | 0.007 | 0.872 | 0.861 |
| ViTL16 | $U$-ensembles | 50 | 20 | 0.254 | 0.045 | 0.007 | 0.871 | 0.871 |
| ViTL16 | $U$-ensembles | 50 | 50 | 0.26 | 0.06 | 0.007 | 0.87 | 0.86 |
| ViTL16 | $U$-ensembles | 50 | 70 | 0.262 | 0.031 | 0.011 | 0.872 | 0.837 |
| ViTL16 | $U$-ensembles | 50 | 100 | 0.258 | 0.019 | 0.01 | 0.868 | 0.855 |
| ViTL16 | $U$-ensembles | 70 | 10 | 0.341 | 0.041 | 0.006 | 0.791 | 0.872 |
| ViTL16 | $U$-ensembles | 70 | 20 | 0.345 | 0.026 | 0.006 | 0.785 | 0.876 |
| ViTL16 | $U$-ensembles | 70 | 50 | 0.344 | 0.03 | 0.006 | 0.784 | 0.878 |
| ViTL16 | $U$-ensembles | 70 | 70 | 0.345 | 0.033 | 0.006 | 0.787 | 0.873 |
| ViTL16 | $U$-ensembles | 70 | 100 | 0.338 | 0.036 | 0.006 | 0.791 | 0.877 |

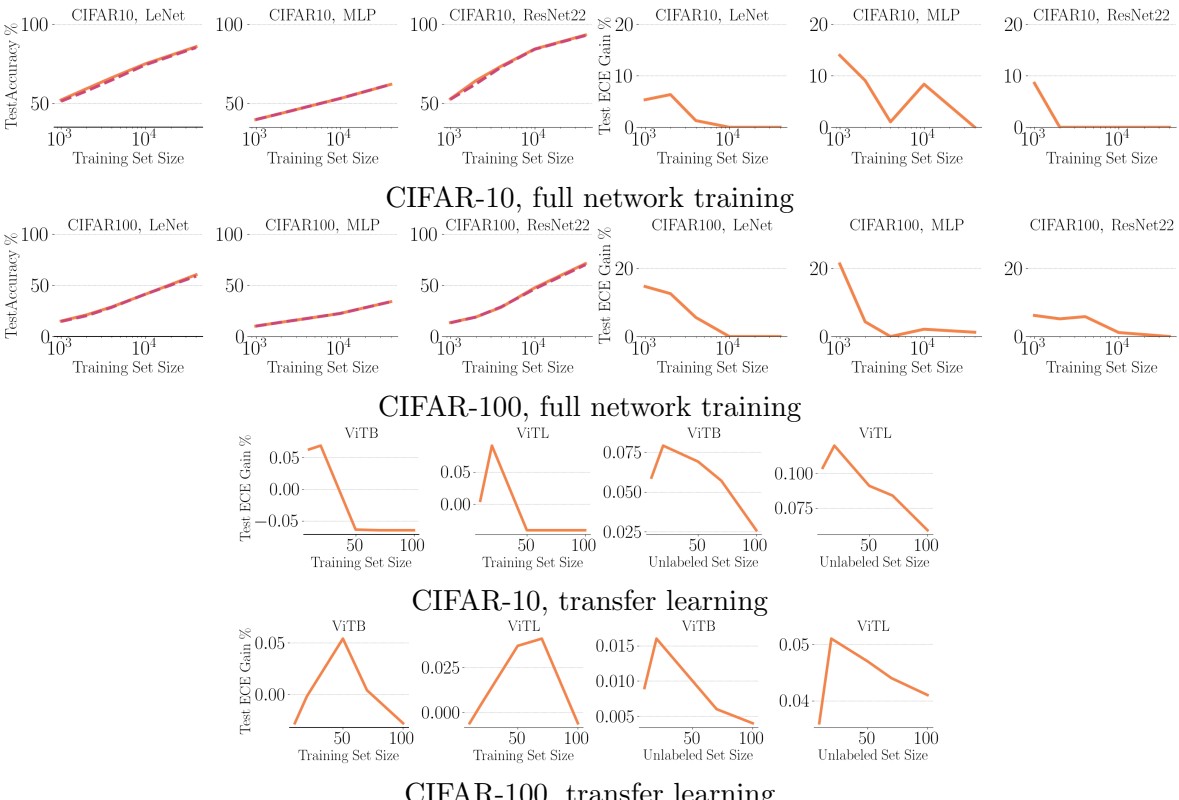

CIFAR-10, full network training

CIFAR-100, full network training

CIFAR-10, transfer learning

CIFAR-100, transfer learning

Figure 7: **Varying the training and unlabeled set.** (First two rows) For both standard and $U$-ensembles, and full network training, we vary the size of the training set $Z$ in $\{1000, 2000, 4000, 10000, 40000\}$. $U$-ensembles have the same test accuracy as standard ensembles (they overlap in the figure) while improving the Expected Calibration Error (ECE). Gains slowly decrease as the training set increases. (Third row) In the ViT transfer learning setup for CIFAR-10 we alternate between fixing the training set $|Z| = 70$ and the unlabeled set $|Z| = 100$ and varying the other set. In both cases calibration first improves and then worsens as each set becomes too large. (Fourth) In the ViT transfer learning setup for CIFAR-100 we alternate between fixing the training set $|Z| = 20$ and the unlabeled set $|Z| = 50$ and varying the other set. In both cases calibration first improves and then worsens as each set becomes too large.

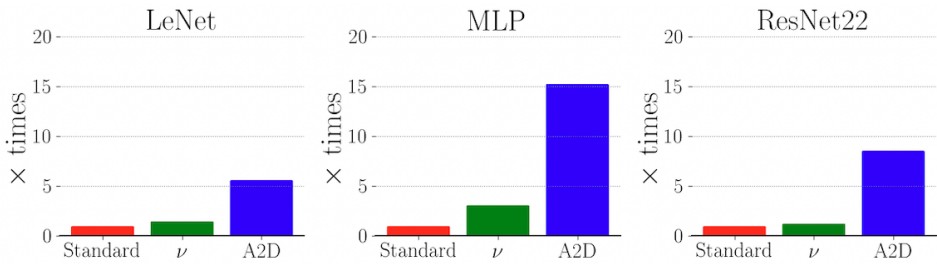

Figure 8: **Comparing the computation times for CIFAR-10 and various architectures.** We compare the training time of Standard, $U$ and Agree to Disagree ensembles, for the CIFAR-10 dataset with 1000 training samples and 5000 unlabeled samples. We plot (total training time)/(epochs * ensemble size). Agree to Disagree ensembles have to be trained sequentially and have higher computational complexity for each member.

