# OpenReview forum: "$U$-ensembles: Improved diversity in the small data regime using unlabeled data"
_approximateinference.org/AABI/2025/Proceedings_Track — AABI 2025 Proceedings Track_

### Official Review · Reviewer_Fvcu · 2025-02-25

**Rating:** 5
**Confidence:** 4

**Review:**

This paper introduces a novel approach called U-ensembles, aimed at diversifying model members for deep ensemble methods. Given an unlabeled dataset, the approach assigns a different randomly selected label to each data point for each ensemble member. The authors provide both theoretical proofs and experimental evidence to demonstrate that U-ensembles outperform standard approaches in terms of uncertainty calibration. A key advantage of this method is its simplicity and ease of implementation. However, there are several limitations:

1. All experiments were conducted using only toy datasets, namely CIFAR-10 and CIFAR-100, with small-scale DNN architectures such as LeNet, ResNet, and MLP.
2. Only basic benchmark methods were included in the experiments, and more advanced ensemble methods were not considered.
3. In some cases, the performance gain in uncertainty calibration comes at the cost of a reduction in accuracy.

Given these issues, it is difficult to assess the practical significance of the approach.

---

### Official Review · Reviewer_yY3A · 2025-02-25
**Nice contribution on ensemble learning for improving model calibration**

**Rating:** 7
**Confidence:** 3

**Review:**

Summary: The paper discusses a deep ensamble approach called U-ensembles that uses unlabeled data during training to improve model calibration. The idea is simple and easy to implement. The paper contains theoretical and empirical results.

Strengths:

- The method is well justified from a theoretical and experimental perspective. Experiments were performed on CIFAR-10 and CIFAR-100 using different architectures (LeNet, MLP, ResNet22, and ViTB16). Experimental results are interesting but not groundbreaking.
- The submission is well-written, feels mostly polished, and easy to understand.

Things to improve:
- There are some errors in the notation: $\mathcal{O} (10^3)$ and $\mathcal{O} (10)$ are, technical speaking, the same. Also, 	the notation $|Z| = {10, 20, 50, 70, 100}$ is not so nice. The notation $1 : K$ is also non-standard and it would be good to quickly introduce this notation.
- There are many paragraphs. For example on page 4 almost each sentence is its own paragraph which hinders the reading flow.
- Bold numbers missing in Table 3.

Overall this is a solid contribution to the workshop proceedings. Some very minor things (see above) can be improved but this could easily be done in the camera-ready submission.

---

### Official Review · Reviewer_Q882 · 2025-02-27
**U-ensembles: Improved diversity in the small data regime using unlabeled data**

**Rating:** 7
**Confidence:** 4

**Review:**

The research developed a novel deep learning ensemble model call U-ensemble. The method achieved improved calibration with minimal changes to the standard deep ensemble pipeline in the presence of unlabelled data.
Though the work was well presented but with some shortcomings:
1. between line 117 and 118, equation was not labelled
2. In line 173, there is proposition 2 without proposition 1
3. Algorithm 1 was presented in a tabular form but not captured as a table
4. In line 175, when defining variables in an equation, equal to (=) should not be used.

---

### Official Review · Reviewer_Sm5U · 2025-02-28

**Rating:** 5
**Confidence:** 3

**Review:**

The paper proposes an ensemble learning strategy where the ensemble fits to training data and to random labels elsewhere. This is an interesting idea, but the paper doesn’t justify clearly why this is done, or contextualise it well. Little conceptual insight is given about the idea. The approach then feels adhoc. The paper does present a PAC-bound on the obtained new loss, which is an interesting result.

A similar idea is explored in other contexts. In Cox point processes the intensity function is fitted by assuming that non-observed regions have zero intensity; while in EBM contrastive learning we assume that density is minimized outside observations. In these domains these ideas are rigorously justified and derived, which I feel is lacking here.

The paper is missing earlier works on repulsive ensembles, which tackle this exact problem. The “Repulsive deep ensembles are Bayesian” paper proposes output diversification, which does almost exactly what this paper also claims to do. Similarly, the “Input-gradient space particle inference for neural network ensembles” proposes diversifying the interpolation (just like in Fig 1). Given that there is an entire subfield of study dedicated to the same problem, I think this submission needs to contextualise and compare itself wrt them.

Furthermore, the U-ensemble can be interpreted as a functional Rademacher prior. There is considerable literature on BNNs with functional priors, which the paper should acknowledge.

Figs 1+2 lack annotations. I’m not sure how to interpret these plots.

The results show consistent and good performance on CIFAR-10 and CIFAR-100: the new ensembles have better calibration than regular ones. Here the key problem is omission of competing methods: (i) BNN-base approaches, (ii) repulsive ensembles, (iii) other semi-supervised neural networks. Furthermore, the results use only small-scale CIFAR, and ancient neural networks. The results have little real-world relevance for this.

Despite the limitations, the idea is clever and likely interesting for the AABI community. I'm rating this as borderline.

---

### Official Review · Reviewer_uoLy · 2025-02-28
**A simple idea that seems to work well**

**Rating:** 6
**Confidence:** 2

**Review:**

The paper presents the "U-ensembles" method to improve the calibration of deep ensembles in small regimes and the presence of unlabeled data. The "U-ensembles" method is claimed to provide diversity, hence better generalization, by inputting the unknown labels randomly and assigning those random imputations to ensembles. Through a PAC-Bayes analysis, the authors prove their method to be diverse and well-calibrated to test data (Theorem 1). Another advantage of the method is its computational complexity as it is embarrassingly parallel. The method is also shown to yield better results in terms of accuracy.

U-ensembles relies on the assumption that the ensemble can perfectly fit random labels. The authors back this assumption with Zhang et al (2021). However, the cited study is empirical. Therefore, it is not very clear how generally this assumption holds.

It is not straightforward to understand the intuition behind filling in the unknown labels 'just randomly'. Diversity is indeed expected to increase but doesn't this randomization make the algorithm behave more in an 'average' manner rather than 'listening' to the data? Why should assigning random labels be better than simply ignoring those examples without the label, even better, somehow predicting those labels (may be with less confidence to allow for generalization)?

- There is a typo in Algorithm 1's title: "with of without".

---

### Official Review · Reviewer_tsJF · 2025-03-02

**Rating:** 6
**Confidence:** 3

**Review:**

The paper proposes a new deep ensemble method U-ensembles, which uses random labels of unlabeled data to enhance the diversity of the ensemble, thereby improving the calibration performance and generalization ability of the model. The paper provides systematic experimental verification with theoretical support from the PAC-Bayesian generalization community and evaluates it on CIFAR-10, CIFAR-100, data perturbation, transfer learning, and out-of-distribution generalization tasks.
Although the paper shows innovation and experimental adequacy in many aspects, there are still some key issues that need further discussion and clarification. Here are my specific review comments:

1. Hyperparameter sensitivity analysis: The paper does not discuss in detail the impact of hyperparameters (such as the loss weight β of unlabeled data) on the model effect.

2. OOD experiments: The paper tests image corruption methods. However, these are still based on the original data distribution. It is recommended to further test completely random noise (such as Gaussian noise) to see if U-ensembles can still improve calibration ability. If it still works, it means that U-ensembles may have a lower dependence on the data itself, which makes the method more valuable.

3. Transfer Learning: The paper's transfer learning experiment was only fine-tuned on CIFAR-10 and CIFAR-100, but these datasets are relatively similar and still in the same field. It is recommended to test cross-domain OOD transfer learning tasks, such as migrating from ImageNet to medical imaging, to verify the generalization ability of the method in different fields.

4. Generalization Analysis: The paper only conducted experiments on CIFAR-10 and CIFAR-100 datasets, mainly for image tasks. It is recommended to expand the test, such as trying U-ensembles on non-image tasks such as text classification to see if it is still effective. If the method can be applied to more tasks, it will be more practical.

5. Necessity of false labels: The core idea of ​​the paper is to generate random labels from unlabeled data to improve the diversity of the integration. For example, if there are only 2 sample classes, is it also valid to randomly expand to K different classes?

6. Impact of complex models (such as ViT): How does U-ensembles perform when complex models (such as ViT) use big data? In Figure 3, it is mainly small CNNs (such as LeNet) and MLPs, rather than complex models (ViT), or the ratio of labeled data to unlabeled data that affects the performance of the method.


Overall, and the experimental design is adequate. However, the theoretical part is quite restrictive, which may limit the scope of application of the method. However, the derivation process is reasonable and there are no obvious problems.

---

### Meta-Review · Area_Chair_APKW · 2025-03-17

**Recommendation:** Accept
**Confidence:** 3

**Metareview:**

The paper present a new method called U-ensembles which helps performance in the small data regime. The reviews were borderline, overall leaning acceptance, remarking on points for improvement such as experimental design (real world use cases etc.) I think the approach is interesting and recommend acceptance, but with the requirement that all suggested points for improvement raised by the reviewers are implemented for camera-ready.

---

### Decision · Program_Chairs · 2025-03-18

**Decision:**

Accept

**Comment:**

Accept conditional on promised revisions in the camera ready.